# OBJECT-BASED SUB-ENVIRONMENT RECOGNITION

## ABSTRACT

Deep learning agents are advancing beyond laboratory settings into the open and realistic environments driven by developments in AI technologies. Since these environments consist of unique sub-environments, empirical recognition of such sub-environments that form the entire environment is essential. Through sub-environment recognition, the agent can 1) retrieve relevant sub-environments for a query, 2) track changes in its circumstances over time and space, and 3) identify similarities between different sub-environments while solving its tasks. To this end, we propose the Object-Based Sub-Environment Recognition (OBSER) framework, a novel Bayesian framework for measuring object-environment and environment-environment relationships using a feature extractor trained with metric learning. We first design the $(\epsilon, \delta)$ Statistically Separable (EDS) function to evaluate to show the robustness of trained representations both theoretically and empirically that the optimized feature extractor can guarantee the precision of the proposed measures. We validate the efficacy of the OBSER framework in open-world and photorealistic environments. The result highlights the strong generalization capability and efficient inference of the proposed framework.

## 1 INTRODUCTION

> *Bright sunlight awakens **the agent**. Bathed in the sunlight filtering through the leaves, it follows its inner voice to complete its tasks. As tasks become familiar, it leaves its spot and walks across the plains. Realizing that it needs **wood**, it remembers that it was in a **forest** full of trees. But since it has come too far to return, it sets out to find a new **forest**.*

Deep learning agents are rapidly proliferating in the broader world, steered by advancements in artificial intelligence. In recent studies, large-scale deep learning models have allowed agents to explore a wider world beyond laboratory settings, encompassing diverse regional characteristics (Shah et al., 2023; Sridhar et al., 2024). Figure 1 illustrates the scenario described in the above quote. In this scenario, where the environment consists of multiple sub-environments, each with unique objects and characteristics, the empirical recognition of these sub-environments plays a key role in performing tasks effectively. Through sub-environment recognition, the agents can 1) retrieve appropriate sub-environments for a given query, 2) measure changes in circumstances over time and space, and 3) infer similarities among them.

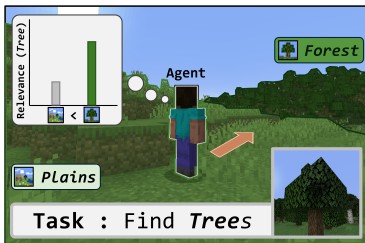

Figure 1: The agent leaves **plains** to reach **forest** because **forest** is a more relevant sub-environment than **plains** to find the trees.

We define sub-environment recognition as an inductive process that interprets a given environment based on the distribution of task-related objects. Previous methods of environment recognition (Barros et al., 2021), such as visual place recognition (Larsson et al., 2019), focus primarily on inferring the agent's location from the visual features of given observations. These methodologies have enabled sophisticated navigation, but there are challenges in making composite and generalized inferences at the sub-environment level to solve the task. In this context, we claim that sub-environment recognition can be achieved through Bayesian inference utilizing the empirical distribution of the task-aware representations of observed objects. Figure 2 shows the inference process with sub-environment recognition. When the 1) query is given as observations, the agent 2) retrieves related

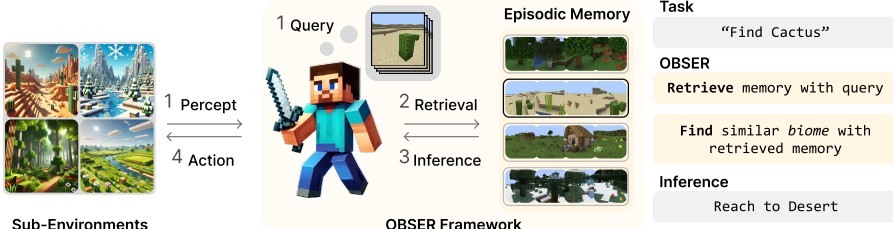

Figure 2: Illustration of the OBSER framework. Upon receiving a query in the form of observations, the agent retrieves related memories from episodic memory and infers the result through sub-environment recognition.

memories from the episodic memory, 3) infers the most appropriate result, and 4) solves the task. Sub-environment recognition is essential for each process to quantify the relationships between objects and environments.

Sub-environment recognition is achieved by inference of three fundamental relationships: object-object, object-environment, and environment-environment relationships. The object-object relationship can be inferred by using similarity that reflects task-relevant information about the object. Metric learning (Khosla et al., 2020; Chen et al., 2021; 2020a; Oquab et al., 2023) is a method that directly optimizes object-object relationships and is known for its robustness in dealing with unstructured data in real-world problems (Cao et al., 2019; Liu et al., 2021; Choi et al., 2024). Therefore, we apply the metric learning model as a feature extractor for sub-environment recognition.

Then *how can object-environment and environment-environment relationships be calculated using a metric learning model* trained only on object-object relationships? To address the issue, we reformulate object similarity using a Bayesian framework to compute the kernel density of empirical distributions within sub-environments. We validate the proposed measures by introducing the $(\epsilon, \delta)$ statistically separable (EDS) function, which represents *separability* and *concentration* of representations, and we theoretically show that when the model satisfies high concentration and separability, the proposed measures can effectively approximate the exact values with explicit class information.

In this paper, we propose an Object-Based Sub-Environment Recognition (OBSER) framework that measures three fundamental relationships with given queries. To validate the OBSER framework, we first validate the proposed measures with the $(\epsilon, \delta)$ values of metric learning and self-supervised learning models in the artificially generated environment using ImageNet dataset (Deng et al., 2009). And we apply the OBSER framework to the Minecraft environment, which is widely used for testbed towards realistic environments (Baker et al., 2022; Qin et al., 2024; Li et al., 2024; Chen et al., 2024), to demonstrate the effectiveness of the proposed framework. We build a dataset with various object observations from different angles from multiple biomes for training baseline models and evaluate the models with the designed miniature environment. We also validate the framework in Replica (Straub et al., 2019) environments by solving the object retrieval task with chained sub-environment recognition.

Our contributions in this paper are summarized as follows:

- **Object-Based Sub-Environment Recognition (OBSER) framework.** We propose the OBSER framework, which can infer three fundamental relationships using an episodic memory: object-object, object-environment, and environment-environment relationship. We present Bayesian metrics for each relationship and demonstrate their effective approximation with multiple sub-environments.

- **$(\epsilon, \delta)$ statistically separable (EDS) function.** We introduce the EDS function $\mathcal{F}_{\epsilon, \delta}$ that measures the separability and concentration of representations extracted from a metric learning model. We prove that high $\delta$ and low $\epsilon$ guarantee that the measures of the OBSER framework converge to the exact value with class information.

- **Validation with multiple sub-environments.** We demonstrate the OBSER framework in the Minecraft and Replica environment, which contain different sub-environments. The proposed framework succesfully infers object retrieval tasks without supervisions via sub-environment recognition with given queries.

## 2 RELATED WORK

**Environment recognition**   Environment recognition is essential for embodied agents to successfully perform tasks such as navigation (Feriol et al., 2020). Previous research has primarily focused on understanding environments by analyzing the semantic distance between current and target observations (Yokoyama et al., 2023; Shah et al., 2023; Sridhar et al., 2024) or by generating scene and environment graphs (Wang et al., 2023) based on trained models. In contrast, we propose a novel approach that defines environmental relationships through the empirical distribution of occurring objects, providing a new method for understanding environments.

**Metric learning**   Metric learning (Sohn, 2016; Liu et al., 2017; Khosla et al., 2020) is a method that optimizes the metric between objects on latent space to reflect the object-object relationships. Self-supervised learning (SSL) is introduced to replace an *oracle* data provider in metric learning, which selects the positive sample, to data augmentation. We use SupCon (Khosla et al., 2020) as metric learning model and MoCo variants (Chen et al., 2020b; 2021), SimCLR (Chen et al., 2020a), and DINO (Caron et al., 2021; Oquab et al., 2023) as SSL models to evaluate the proposed measures.

**Kernel method**   Kernel density estimation is a non-parametric method for estimating the measures such as probabilistic density function (Zhang et al., 2018; Ghosh et al., 2006) and Kullback-Liebler divergence (Ahuja, 2019; Ghimire et al., 2021). We utilize the object similarity as a kernel to approximate the distribution of sub-environments. We also validate the precision of the measures with EDS function, which computes the kernel density accumulated with class-wise distribution.

## 3 DEFINITION OF SUB-ENVIRONMENT RECOGNITION

### 3.1 SUB-ENVIRONMENT AS AN OBJECT DISTRIBUTION

In this section, we introduce the concept of an environment consisting of multiple sub-environments, along with measures for Bayesian inference. Suppose that an object $x$ is defined in a domain $x \in \mathcal{X}$, which is a Borel set. A *latent* class $c \in \mathcal{C}$ indicates the degree of generalization required to solve the task $\mathcal{T}$ (Arora et al., 2019; Ash et al., 2021; Awasthi et al., 2022). With latent classes, $\mathcal{X}$ is partitioned into $\mathcal{X}_c$, which satisfies $\mathcal{X} = \bigcup_{c \in \mathcal{C}} \mathcal{X}_c, \mathcal{X}_c \cap \mathcal{X}_{c'} = \emptyset$. Consequently, this allows us to consider sub-environments as a mixture of probability distributions, such as $x \sim \mu$ and $x \sim \nu$, with class distributions and class-wise object distributions $\mathcal{D}_c(x) := p(x|c)$:

$$\mu := \sum_{c \in \mathcal{C}} \rho_c^{(\mu)} \cdot \mathcal{D}_c, \quad \nu := \sum_{c \in \mathcal{C}} \rho_c^{(\nu)} \cdot \mathcal{D}_c,$$

$$\forall c' \neq c \in \mathcal{C}, \forall x \in \mathcal{X}_c, \quad \mathcal{D}_c(x)\mathcal{D}_{c'}(x) = 0.$$

Since there are differences in the occurrence of certain objects between sub-environments, we define $\rho_c^{(\mu)} := p(c; \mu)$ as the *existence probability of the object* with latent class $c$, or the marginal class distribution of a sub-environment $\mu$. We also assume that sub-environments share the same class-wise data distribution $\mathcal{D}_c(x) := p(x|c)$ to focus primarily on the existence probability. Throughout the paper, an environment $\mathcal{E}$ is defined as a set of sub-environments $\mu_s$ associated with the regions $\mathcal{R}_s$: $\mathcal{E} := \{(\mu_s, \mathcal{R}_s)\}_{i=1}^{S}$.

### 3.2 SUB-ENVIRONMENT RECOGNITION

General metric learning models focus on learning the relationships between objects for a given task. Sub-environment recognition, in relation to these models, is a methodology for computing the relationships between objects and environments, or between environments, given a task-aware mapping function $\mathcal{F}$. With a function $\mathcal{F} : \mathcal{X} \to \mathcal{Z}$, which maps the data onto latent space $\mathcal{Z}$, we define three measures for each sub-environment recognition:

i) Object Similarity (obj. − obj.): $\quad p(c = c'|x, x'; \mathcal{F})$

ii) Object Existence Probability (obj. − env.): $\quad \hat{\rho}_{c|x}^{(\mu)} := p(c|x \in \mathcal{X}_c; \mu, \mathcal{F})$

iii) KL Divergence (env. − env.): $\quad \hat{D}_{\mathrm{KL}}(\mu||\nu; \mathcal{F}).$

The object-object relationship is used to retrieve the most appropriate object that shares the same latent class with the given query object. With the object-environment relationship, an agent can reach a sub-environment that contains queried objects with the highest probability. The environment-environment relationship allows an agent to measure the changes in the environment and infer the similarity between sub-environments. An agent can understand the environment through sub-environment recognition and perform effective inference and exploration. This means that accurately approximating the measures of sub-environment recognition is essential for improving the inference accuracy.

## 3.3 KERNEL DENSITY ESTIMATION

To describe a metric between the representations of two objects $x$ and $x'$ for probabilistic inference, we define a kernel $\mathcal{K}_\mathcal{Z}(x, x'; \mathcal{F})$ as the belief of sharing the same latent class. With a metric function $d_\mathcal{Z} : \mathcal{Z} \times \mathcal{Z} \to \mathbb{R}_0^+$ and a kernel function $h : \mathbb{R}_0^+ \to [0, 1]$, $\mathcal{K}_\mathcal{Z}(x, x'; \mathcal{F})$ is defined as:

$$x, x' \in \mathcal{X}, \quad \mathcal{K}_\mathcal{Z}(x, x'; \mathcal{F}) = p(c = c'|x, x'; \mathcal{F}) := h(d_\mathcal{Z}(\mathcal{F}(x), \mathcal{F}(x'))). \tag{1}$$

With the kernel function, we can compute the kernel density of $x$ by accumulating the kernel with respect to the given distribution $\mu$.

$$\mathbb{E}_{x' \sim \mu} [\mathcal{K}_\mathcal{Z}(x, x'; \mathcal{F})] = \mathbb{E}_{x' \sim \mu} [h(d_\mathcal{Z}(\mathcal{F}(x), \mathcal{F}(x')))] \tag{2}$$

Kernel density implies the expected relationship between queried data $x$ and distribution $\mu$. To apply kernel-based methods for measurements, the alignment of representations extracted with the feature extractor is important. In this context, we introduce $(\epsilon, \delta)$ statistically separable function $\mathcal{F}_{\epsilon,\delta}$, which represents the property of the feature extractor with kernel densities.

# 4 $(\epsilon, \delta)$ STATISTICALLY SEPARABLE (EDS) FUNCTION $\mathcal{F}_{\epsilon,\delta}$

In this section, we define the $(\epsilon, \delta)$ statistically separable function which determines two essential properties of the feature extractor: $\epsilon$ for *separability* and $\delta$ for *concentration*. Figure 3 shows the visual concept of two properties in the embedding space. *Separability* indicates that the metric between data from different latent classes should be larger. On the other hand, *concentration* indicates that the metric between data from the same latent class should be relatively small.

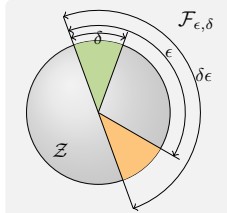

Figure 3: EDS function.

## 4.1 $(\epsilon, \delta)$ STATISTICALLY SEPARABLE (EDS) FUNCTION $\mathcal{F}_{\epsilon,\delta}$

First, we introduce the $(\epsilon, \delta)$ statistically separable (EDS) function. For all objects, $x$ with latent class $c \in \mathcal{C}$, delta ($\delta$) and epsilon ($\epsilon$) are defined with the kernel densities with the distribution $\mathcal{D}_c$ with the same class, and the distributions $\mathcal{D}_{c'}$ with different classes, respectively.

**Definition 1** ($(\epsilon, \delta)$ statistically separable function). *A function $\mathcal{F}_{\epsilon,\delta}$ is $(\epsilon, \delta)$ statistically separable if it satisfies the following in $\mu$-almost everwhere with $\exists \epsilon, \delta, 0 \le \epsilon \le \delta \le 1$:*

$$\forall c \in \mathcal{C}, \ x \in \mathcal{X}_c, \quad \delta \le \mathbb{E}_{x' \sim \mathcal{D}_c} [\mathcal{K}_\mathcal{Z}(x, x'; \mathcal{F}_{\epsilon,\delta})] \le 1, \tag{3}$$

$$\forall c' \ne c, \quad \delta\epsilon \le \mathbb{E}_{x' \sim \mathcal{D}_{c'}} [\mathcal{K}_\mathcal{Z}(x, x'; \mathcal{F}_{\epsilon,\delta})] \le \epsilon. \tag{4}$$

Intuitively, by definition, note that smaller $\epsilon$ and larger $\delta$ imply that the feature extractor is more robust in downstream tasks. In other words, with the optimal EDS function, which has $\delta = 1$ and $\epsilon = 0$, the latent class $c$ of the data $x$ is directly derived, and the joint distribution $p(c, x; \mu, \mathcal{F}_{\epsilon,\delta})$ becomes equivalent to the ground-truth joint distribution $p(c, x; \mu)$. By analyzing the optimization process of the EDS function, we can understand the dynamics of the EDS function.

## 4.2 OPTIMIZATION OF THE EDS FUNCTION

We define the optimization of the EDS function as fitting the estimated joint distribution $p(c, x; \mu, \mathcal{F}_{\epsilon,\delta})$ to the ground-truth joint distribution $p(c, x; \mu)$ inspired by the idea of Choi et al.

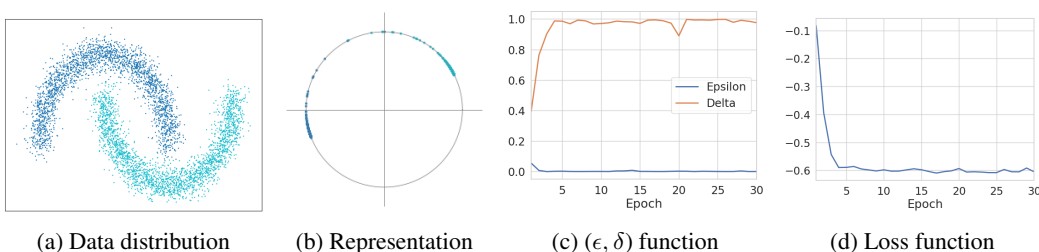

(a) Data distribution    (b) Representation    (c) $(\epsilon, \delta)$ function    (d) Loss function

Figure 4: Visualization of the optimization of the EDS function with Moons dataset.

(2024). Thus, we formulate the optimization problem of the EDS function as minimizing the KL divergence between $p(c, x; \mu, \mathcal{F}_{\epsilon,\delta})$ and $p(c, x; \mu)$. By tightening the bound of KL divergence, the EDS function is optimized in terms of both separability and concentration properties.

**Theorem 1** (Optimization of $\mathcal{F}_{\epsilon,\delta}$ via generalized metric learning). *For an EDS function $\mathcal{F}_{\epsilon,\delta}$, let $\exists k \geq 1, \delta = k \cdot \epsilon$. The upperbound of $D_{KL}(p(c, x; \mu) || p(c, x; \mu, \mathcal{F}_{\epsilon,\delta}))$ is derived as:*

$$0 \leq D_{KL}(p(c, x; \mu) || p(c, x; \mu, \mathcal{F}_{\epsilon,\delta})) \leq \log\left(1 + \frac{|\mathcal{C}| - 1}{k}\right) := \Delta\mathcal{H}, \qquad (5)$$

*and if $\Delta\mathcal{H} \to +0$, then $k \to \infty$.*

*Sketch of Proof.* We first rederive the KL divergence $D_{\text{KL}}(p(c, x; \mu) || p(c, x; \mu, \mathcal{F}_{\epsilon,\delta}))$, introduced in Choi et al. (2024), with kernel densities. The KL divergence is then rewrited as Equation 6:

$$D_{\text{KL}}(p(c, x; \mu) || p(c, x; \mu, \mathcal{F}_{\epsilon,\delta})) = \mathbb{E}_{x \sim \mu}\left[-\log\frac{\mathbb{E}_{x' \sim \mathcal{D}_+}[\mathcal{K}_{\mathcal{Z}}(x, x'; \mathcal{F}_{\epsilon,\delta})]}{\mathbb{E}_{x' \sim \mu}[\mathcal{K}_{\mathcal{Z}}(x, x'; \mathcal{F}_{\epsilon,\delta})]}\right] + \mathcal{H}(\mathcal{C}; \mu). \qquad (6)$$

We find the upper bound of Equation 6 by applying Definition 1 and apply the condition $\delta = k\epsilon$. Then, $k$ is defined with $\Delta\mathcal{H}$, and we show $\Delta\mathcal{H} \to 0, k \to \infty$. □

Note that the optimization with Equation 6 becomes equivalent to InfoNCE (Oord et al., 2018) by setting the kernel $\mathcal{K}_{\mathcal{Z}}(x, x'; \mathcal{F}_{\epsilon,\delta})$ as follows:

$$\mathcal{K}_{\mathcal{Z}}(x, x'; \mathcal{F}_{\epsilon,\delta}) = \exp\{(\mathcal{F}_{\epsilon,\delta}(x)^\top \mathcal{F}_{\epsilon,\delta}(x') - 1)/\tau\}, \quad \mathcal{K}_{\mathcal{Z}}(x, x'; \mathcal{F}_{\epsilon,\delta}) \in [\exp(-2/\tau), 1], \quad (7)$$

with a temperature parameter $\tau$. Figure 4 shows the visualization of the optimization of $\mathcal{F}_{\epsilon,\delta}$ with the Moons dataset. After the optimization, the trained feature extractor successfully maps the data onto a 2-dimensional sphere. During the optimization, $\epsilon$ converges to 0 while $\delta$ converges to 1 as the training loss is minimized. This implies that the $\epsilon$ and $\delta$ values accurately represent the robustness of arbitrary feature extractors.

## 5 OBJECT-BASED SUB-ENVIRONMENT RECOGNITION (OBSER)

In the previous section, we discussed that the EDS function can indicate the orientation of representations. In this section, each recognition concept is reinterpreted through the lens of the EDS function, and each measure is estimated using an empirical distribution of observations. Furthermore, it is shown both theoretically and practically that the optimized EDS function, with its high concentration and separability, guarantees the accuracy of the estimated measures.

We validate the proposed measures with metric learning or SSL models using the ImageNet dataset. To employ a hypersphere space as an embedding space instead of Euclidean space, we set the kernel $\mathcal{K}_{\mathcal{Z}}(x, x'; \mathcal{F}_{\epsilon,\delta})$ as same as Equation 7. Pre-trained weights for each model are used for reproducibility. More details on the evaluation can be found in Appendix B.2.

### 5.1 OBJECT-OBJECT RECOGNITION: OBJECT SIMILARITY

Object-object recognition is a fundamental and the most important recognition concept for achieving other sub-environment recognition. With the query object $x_q$ and candidate objects $\{x_1, \cdots, x_K\}$,

Table 1: EDS values of pretrained metric learning and SSL models with ImageNet classification accuracies. To verify the reported performance of the pretrained models, the linear probing accuracy is additionally presented. (*: normalized backbone features are utilized instead of embedding vector.)

| Model | Architecture | | EDS ($\tau = 1.0$) | | EDS ($\tau = 0.5$) | | EDS ($\tau = 0.1$) | | Linear | Mean | | | KNN | | |
|---|---|---|---|---|---|---|---|---|---|---|---|---|---|---|---|
| | backbone | dim | del | eps | del | eps | del | eps | 1 | 1 | 3 | 5 | 3 | 5 | 7 |
| MoCo-v2 | ResNet-50 | 128 | 0.450 | 0.390 | 0.219 | 0.154 | 0.025 | 0.001 | 71.1 | 46.68 | 67.62 | 74.92 | 47.03 | 48.90 | 49.55 |
| MoCo-v3 | ViT-S | 256 | 0.442 | 0.391 | 0.208 | 0.155 | 0.023 | 0.000 | 73.2 | 58.35 | 74.31 | 79.07 | 57.00 | 58.47 | 59.05 |
| | ViT-B | 256 | 0.462 | 0.389 | 0.230 | 0.153 | 0.026 | 0.000 | 76.7 | 62.15 | 78.79 | 83.27 | 60.90 | 62.35 | 62.77 |
| SimCLR | ResNet-50 | 128 | 0.427 | 0.401 | 0.194 | 0.163 | 0.021 | 0.000 | 67.8 | 46.55 | 65.76 | 72.69 | 43.63 | 45.73 | 46.86 |
| DINO-v1 (*) | ViT-S | 384 | 0.502 | 0.408 | 0.264 | 0.168 | 0.024 | 0.000 | 79.7 | 71.13 | 86.43 | 90.36 | 73.08 | 74.05 | 74.41 |
| | ViT-B | 768 | 0.500 | 0.410 | 0.260 | 0.169 | 0.023 | 0.000 | 80.1 | 70.15 | 86.38 | 90.56 | 71.62 | 72.66 | 72.87 |
| DINO-v2 (*) | ViT-S | 384 | 0.473 | 0.391 | 0.235 | 0.154 | 0.022 | 0.000 | 81.1 | 71.23 | 87.33 | 91.44 | 73.86 | 74.85 | 75.13 |
| | ViT-B | 768 | 0.475 | 0.384 | 0.238 | 0.148 | 0.023 | 0.000 | **84.5** | 76.63 | **91.08** | **94.16** | 78.15 | 78.95 | 79.20 |
| SupCon | ResNet-50 | 128 | **0.681** | 0.478 | **0.477** | 0.232 | **0.074** | 0.002 | 74.1 | **79.08** | **90.69** | 93.18 | **78.19** | **78.65** | **78.81** |

conventional evaluation metrics for the classification task orginated from the following formula:

$$x_{\text{target}} := \arg\max_{x_k \in \{x_1, \cdot, x_K\}} \mathcal{K}_\mathcal{Z}(x_q, x_k; \mathcal{F}_{\epsilon,\delta}).$$

We choose the mean classifier to show the influence of *separability* and the KNN classifier for *concentration*. Table 1 shows EDS values and classification accuracies of various metric and self-supervised learning (SSL) models with the ImageNet dataset. SupCon, a metric learning model, shows better mean classifier accuracy and KNN accuracy than other SSL models. DINO-v2 shows the most competitive performance among SSL models and the best linear probing accuracy. Considering the EDS values, we can say that a larger gap between $\delta$ and $\epsilon$ is important for the downstream task performance, and models with larger embedding space can perform better when the models have similar EDS values.

## 5.2 OBJECT-ENVIRONMENT RECOGNITION: OBJECT EXISTENCE PROBABILITY

Object-environment recognition is about retrieving the most appropriate sub-environment $\mu$ with a given query object. Given a query $x_q$, it is logical to define the most appropriate sub-environment as the one with the highest existence probability of association with $x_q$. With optimized $\mathcal{F}_{\epsilon,\delta}$, we can estimate the existence probability of association with the kernel density. With samples $\{X_1^{(\mu)}, \cdots, X_N^{(\mu)}\} \sim \mu$ and query $x$, $\exists c \in \mathcal{C}, x \in \mathcal{X}_c$, the estimated object existence probability $\hat{\rho}_{c|x}^{(\mu)}$ can be computed as $\hat{\rho}_{c|x}^{(\mu)} := \mathbb{E}_{x' \sim \mu} [\mathcal{K}_\mathcal{Z}(x, x'; \mathcal{F}_{\epsilon,\delta})]$.

Depending on the $\delta$ value of the EDS function, there is a probability that the representation for query $x$ may be distant from the actual concept of the objects. Therefore, instead of using a single query, we utilize multiple queries $Q = \{x_1, \cdots, x_k\}$ for the corresponding object. In this case, the geodesic mean representation $\bar{r} \simeq \log \sum_q \exp(-1/2 \cdot d_\mathcal{Z}^2(\mathcal{F}_{\epsilon,\delta}(x_q), \bar{r}))$ is used to compute the kernel density. By introducing an artificial query $\bar{x}$ which satisfies $\mathcal{F}_{\epsilon,\delta}(\bar{x}) = \bar{r}$, the estimated object existence probability is rewritten as $\hat{\rho}_{c|\bar{x}}^{(\mu)}$.

**Definition 2** (Existence probability estimation). *With samples $\{X_1^{(\mu)}, \cdots, X_N^{(\mu)}\} \sim \mu$ and query $Q = \{x_1, \cdots, x_k\}$, $\exists c \in \mathcal{C}, \forall q \in \{1, \cdots, k\}, x_q \in \mathcal{X}_c$, suppose that there exists $\bar{x}$ which satisfies $\mathcal{F}_{\epsilon,\delta}(\bar{x}) = \bar{r}$, $\bar{r} \simeq \log \sum_q \exp(-1/2 \cdot d_\mathcal{Z}^2(\mathcal{F}_{\epsilon,\delta}(x_q), \bar{r}))$. The estimated object existence probability $\hat{\rho}_{c|\bar{x}}^{(\mu)}$ can be computed as:*

$$\hat{\rho}_{c|\bar{x}}^{(\mu)} := \mathbb{E}_{x' \sim \mu} [\mathcal{K}_\mathcal{Z}(\bar{x}, x'; \mathcal{F}_{\epsilon,\delta})] \simeq \frac{1}{N} \sum_{i=1}^{N} \mathcal{K}_\mathcal{Z}(\bar{x}, X_i^{(\mu)}; \mathcal{F}_{\epsilon,\delta}). \tag{8}$$

Figure 5 shows the estimated object existence probability with artificially generated subsets of ImageNet. A Zipf distribution (Manning, 1999; Joseph et al., 2021), a long-tailed distribution, with $\rho_c^{(\mu)} \propto 1/c^{-\alpha}, \alpha = 0.5$ is used to demonstrate the real-world environment. At low temperatures, SSL models with relatively low $\delta$ tend to underestimate probabilities, while at high temperatures, due to the high $\epsilon$ values, all models tend to overestimate probabilities. To address this issue, we

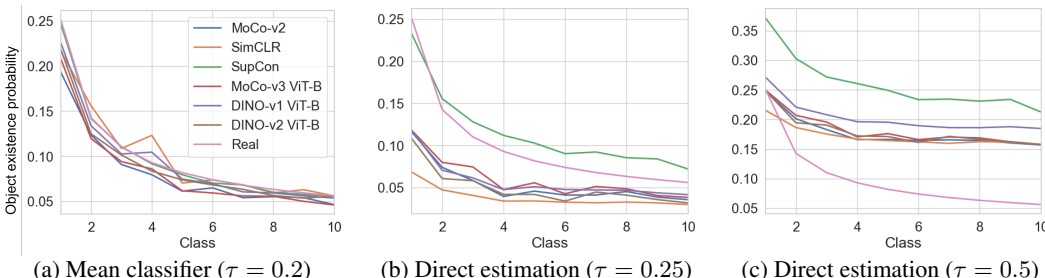

(a) Mean classifier ($\tau = 0.2$)  (b) Direct estimation ($\tau = 0.25$)  (c) Direct estimation ($\tau = 0.5$)

Figure 5: Visualization of object existence probability estimation. Direct estimation leads to under-estimation ($\tau = 0.25$) or overestimation ($\tau = 0.5$). To solve the problem, we use geodesic mean vector $\bar{r}$ as a mean classifier with an adaptive threshold. (Mean values with 5 different seeds.)

use the mean vector of the query $\bar{r}$ as a classifier, quantifying the kernel value with a threshold proportional to the kernel density with $\bar{r}$. By the quantization, the precision of the estimation is dramatically improved. Appendix C.2 shows more analyses for this experiment.

## 5.3 Environment-environment recognition: KL Divergence

To define the difference between two sub-environments, we can utilize a metric between distributions from each sub-environment. The Kullback-Leibler (KL) divergence is one of the most common measures between two distributions $\mu$ and $\nu$. Under the assumption of the data distribution in Section 3, the KL divergence between $\mu$ and $\nu$ is derived with the corresponding class distributions:

$$D_{\mathrm{KL}}(\mu||\nu) := D_{\mathrm{KL}}(p(c,x;\mu)||p(c,x;\nu)) = \sum_{c \in \mathcal{C}} \rho_c^{(\mu)} \cdot \log \frac{\rho_c^{(\mu)}}{\rho_c^{(\nu)}}. \tag{9}$$

Since the agent cannot access all the class information of each observation without supervision, we use kernel density estimation to approximate the KL divergence, denoted as $\hat{D}_{\mathrm{KL}}(\mu||\nu; \mathcal{F}_{\epsilon,\delta})$.

**Definition 3** (KL divergence estimation). *With given samples $\{X_1^{(\mu)}, \cdots, X_N^{(\mu)}\} \sim \mu$ and $\{X_1^{(\nu)}, \cdots, X_M^{(\nu)}\} \sim \nu$, approximated KL divergence $\hat{D}_{KL}(\mu||\nu; \mathcal{F}_{\epsilon,\delta})$ can be computed as:*

$$\hat{D}_{KL}(\mu||\nu; \mathcal{F}_{\epsilon,\delta}) := \mathbb{E}_{x \sim \mu} \left[ \log \frac{\mathbb{E}_{x' \sim \mu}\left[\mathcal{K}_{\mathcal{Z}}(x, x'; \mathcal{F}_{\epsilon,\delta})\right]}{\mathbb{E}_{x' \sim \nu}\left[\mathcal{K}_{\mathcal{Z}}(x, x'; \mathcal{F}_{\epsilon,\delta})\right]} \right]$$

$$\simeq \frac{1}{N} \sum_{i=1}^{N} \left( \log \sum_{j=1}^{N} \mathcal{K}_{\mathcal{Z}}(X_i^{(\mu)}, X_j^{(\mu)}; \mathcal{F}_{\epsilon,\delta}) - \log \sum_{j=1}^{M} \mathcal{K}_{\mathcal{Z}}(X_i^{(\mu)}, X_j^{(\nu)}; \mathcal{F}_{\epsilon,\delta}) + \log \frac{M}{N} \right). \tag{10}$$

To validate the proposed measure in Definition 3, we show that the divergence $\hat{D}_{\mathrm{KL}}(\mu||\nu; \mathcal{F})$ converges to $D_{\mathrm{KL}}(\mu||\nu)$ with the optimized EDS function.

**Theorem 2** (KL divergence with $\mathcal{F}_{\epsilon,\delta}$). *For an EDS function $\mathcal{F}_{\epsilon,\delta}$, the proposed measure in Definition 3 has the bound of:*

$$\left| \hat{D}_{KL}(\mu||\nu; \mathcal{F}_{\epsilon,\delta}) - \sum_{c \in \mathcal{C}} \rho_c^{(\mu)} \cdot \log \left( \frac{\rho_c^{(\mu)} + (1 - \rho_c^{(\mu)}) \cdot \epsilon}{\rho_c^{(\nu)} + (1 - \rho_c^{(\nu)}) \cdot \epsilon} \right) \right| \leq -\log \delta, \tag{11}$$

*in $(\mu, \nu)$-almost everywhere. With optimized $\mathcal{F}_{\epsilon,\delta}$, such that $\delta \to 1, \epsilon \to 0$, $\hat{D}_{KL}(\mu||\nu; \mathcal{F}_{\epsilon,\delta})$ converges to $D_{KL}(\mu||\nu)$.*

With Theorem 2, we find that the approximated KL divergence converges to the exact value when $\epsilon \to 0$ and $\delta \to 1$ are satisfied. In other words, by optimizing $\mathcal{F}_{\epsilon,\delta}$ via Theorem 1 in a compact space, the error bound becomes tighter, and eventually the measure is squeezed to $D_{\mathrm{KL}}(\mu||\nu)$.

In terms of the kernel density estimation, a large $\epsilon$ causes the *oversmoothing* effect, which makes the KL divergence converge to 0. On the other hand, a small $\delta$ causes the *fragmentation* effect on

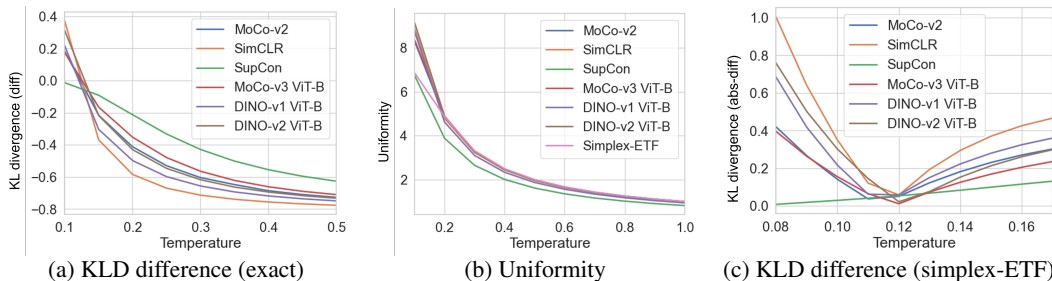

Figure 6: Visualization of KL divergence estimation. Kernel density estimation via Definition 3 can cause *fragmentation* with small $\tau$, and *oversmoothing* with large $\tau$. Empirically, we show that measures with $\tau$ near 0.12 have the best precision. (Mean values with 5 different seeds.)

clusters, which makes the overestimation of the KL divergence. To solve this trade-off, we claim that the temperature $\tau$ plays a role by adjusting the amount of effect caused by both values.

Figure 6 shows the quantitative results with a scenario generated with the ImageNet dataset. We add a plot of the ideal condition, denoted as simplex-ETF (Equiangular Tight Frame), which has been known as the optimal alignment of representations in both supervised learning and metric learning (Papyan et al., 2020; Awasthi et al., 2022; Li et al., 2022). In this case, we set $\epsilon = \exp(-|\mathcal{C}|/((|\mathcal{C}| - 1) \cdot \tau))$ and $\delta = 1$ for simplex-ETF. Figure 6-(a) shows the difference between the approximated and exact values at different temperatures. All measures become smaller with larger $\epsilon$, which supports the *oversmoothing* effect. In Figure 6-(b), we plot the uniformity measures (Wang & Isola, 2020; Tian et al., 2021; Fang et al., 2024) of the models to show the *fragmentation* effect in smaller $\delta$. In the self-supervised learning models, the uniformity values begin to explode at small $\tau$ below 0.2, and this can explain the overestimation of the KL divergence of these models in small temperatures. We have also empirically found that the $\tau$ near 0.12 performs best in this scenario. Please see Appendix C.2 for more experiments with different scenarios.

## 6 VALIDATION OF OBSER FRAMEWORK IN MULTIPLE SUB-ENVIRONMENTS

### 6.1 OPEN-WORLD ENVIRONMENT (MINECRAFT)

**Minecraft environment**   Minecraft, an open-world sandbox game, is often used as an intermediate environment to reach the realistic environments (Baker et al., 2022; Qin et al., 2024; Li et al., 2024; Chen et al., 2024). In Minecraft, there are various sub-environments from nature, the *biomes*, which have unique objects in each of them. We first choose 10 different biomes and gather ego-centric observations of occurring objects in each biome to build a dataset for training. To train each model, we utilize a dataset of object observations with about 26k observations from 25 object classes. Refer to Appendix B.3 for more details of the dataset.

**Model description**   SimCLR (Chen et al., 2020a), MoCo-v2 (Chen et al., 2020b), and Sup-Con (Khosla et al., 2020) models are used for this experiment. Since the classification task of Minecraft is relatively easier than that of ImageNet, we use ResNet50 (He et al., 2016) as the backbone. The dimension of the embedding space is set to 128, and the temperature $\tau$ is set to 0.2. Each model is trained for 10 epochs as a warm-up. We then train SupCon and SimCLR for 20 epochs and MoCo for 100 epochs due to the slow learning at the beginning of the training.

Unlike conventional vision datasets, we can gather observations of the same object from multiple directions. Thus, the directions of the observations can be used as an additional inductive bias (Pantazis & Salvaris, 2022; Scherr et al., 2022). To train the model, different observations of the same object are chosen as positive samples in both metric learning and SSL algorithms. Additional details about the model training can be found in Appendix B.4.

**Miniature environment and episodic memory**   For evaluation, we have also designed a smaller version of the generated world, a Miniature environment, which contains all biomes used for training. Figure 7-(a) shows the landscape of the Miniature environment. We gather random observations

Table 2: EDS values of metric learning and SSL models with classification accuracies (Minecraft).

| Model | EDS ($\tau = 1.0$) | | EDS ($\tau = 0.5$) | | EDS ($\tau = 0.1$) | | Mean | | | KNN | | |
|---|---|---|---|---|---|---|---|---|---|---|---|---|
| | del | eps | del | eps | del | eps | 1 | 3 | 5 | 3 | 5 | 7 |
| MoCoV2 | 0.768 | 0.395 | 0.648 | 0.163 | 0.376 | 0.005 | 74.74 | 82.77 | 84.10 | 94.08 | 94.04 | 94.04 |
| SimCLR | 0.706 | 0.396 | 0.555 | 0.162 | 0.267 | 0.003 | 76.95 | 83.61 | 84.36 | 95.57 | 95.30 | 95.46 |
| SupCon | **0.893** | 0.365 | **0.850** | 0.136 | **0.782** | 0.005 | **86.49** | **86.56** | **86.67** | **96.83** | **96.83** | **96.83** |

(a) Miniature environment          (b) Example episodic memory

Figure 7: Visualization of Miniature Environment. The Miniature environment consists of 10 different biomes: `snowy_taiga`, `taiga`, `forest`, `snowy_plains`, `plains`, `swamp`, `savanna`, `dark_forest`, `desert` and `jungle`, arranged from top-left to bottom-right.

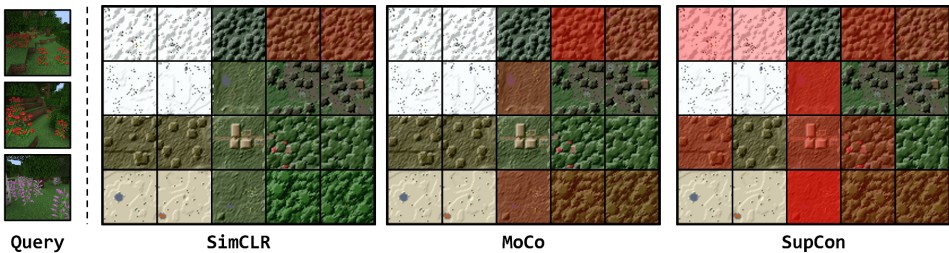

Figure 8: Heatmaps of existent probability with Miniature environment. When observations of `flower` from `forest` are given, SSL models consider the ambient background information of the sub-environment, while the metric learning model focuses on the class information.

from each biome in the map to form episodic memory for evaluating the proposed measures. Figure 7-(b) shows the example episodic memory. For more information on the map and episodic memory, please refer to Appendix B.4.

**Object-Object recognition** We first measure the classification accuracies of each trained model. In Table 2, SupCon performs better than SSL models with the classification task. It also consistently supports our claim regarding the EDS values of the mapping function and task performance.

**Object-Environment recognition** We validate that each model can estimate the object existence probability with the given query objects. Figure 8 shows a heatmap of the existence probability of `flower` for each grid. Note that the query only contains observations of `flower` in `forest`. Self-supervised learning models only focus on `forest` with a high existence probability, unlike SupCon, which estimates the existence probability of `flower` in each biome.

**Environment-Environment recognition** To validate environment-environment recognition, we perform a task to retrieve similar grids by measuring KL divergence using the observations from each grid. Figure 9 shows the results of retrieving the Top-4 grids for four queries with different biomes: `forest`, `desert`, `plains`, and `jungle`. We have observed that all models prioritize retrieving grids with the same biome.

## 6.2 CHAINED INFERENCES IN PHOTOREALISTIC ENVIRONMENT (REPLICA)

To show the proposed framework can be efficiently applied to photorealistic environments, we utilize the Replica environment (Straub et al., 2019), a 3D indoor environment. We first gather random

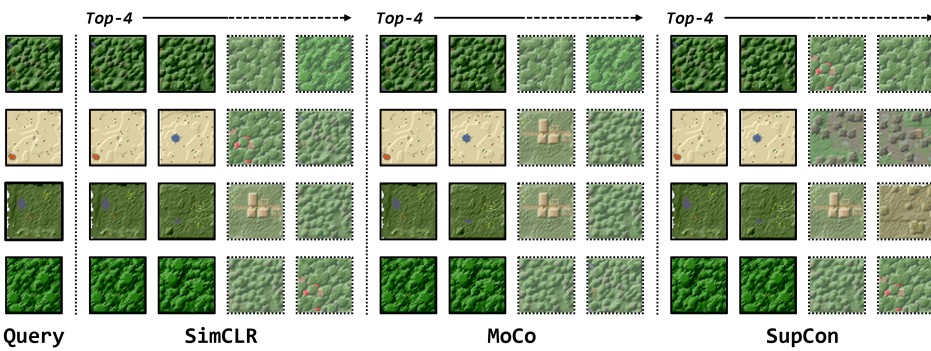

Figure 9: Environment-environment retrieval task with Miniature environment. With the queries of observations from 4 different biomes, `forest`, `desert`, `plains` and `jungle`, both SSL models and the metric learning model successfully retrieve the similar sub-environments.

Table 3: Success rate of object retrieval task with Replica environment. We have queried 10 different objects and marked it as *success* if the correct object is among the Top-5 retrieved objects.

|  | SupCon | SimCLR | MoCo-v2 | MoCo-v3-B | DINO-v1-B | DINO-v2-B |
|---|---|---|---|---|---|---|
| Seen (obj-obj, 48 rooms) | 10/10 | 10/10 | 10/10 | 10/10 | 10/10 | 8/10 |
| Seen (Top-1 room) | 9/10 | 5/10 | 7/10 | 8/10 | **10/10** | **9/10** |
| Seen (Top-3 rooms) | 9/10 | 6/10 | 8/10 | 9/10 | **10/10** | 8/10 |
| Unseen (obj-obj, 35 rooms) | 9/10 | 10/10 | 10/10 | 9/10 | 9/10 | 8/10 |
| Unseen (Top-1 room) | 3/10 | 3/10 | 7/10 | 7/10 | 5/10 | **8/10** |
| Unseen (Top-3 rooms) | 6/10 | 6/10 | 6/10 | **9/10** | 6/10 | 8/10 |

observations from each room and extract object-wise observations with ground-truth segmentations. With observations as episodic memory, we apply the OBSER framework for the object retrieval task with three-step inferences: i) retrieval on memory with a given query, ii) retrieval of similar sub-environment with the memory, and iii) retrieval of similar objects in the room.

We have conducted experiments in two settings: *seen* setting, where the environment and episodic memory are the same, and *unseen* setting, where the environment and episodic memory are exclusive. During sub-environment retrieval in step ii), we have retrieved top-1 and top-3 rooms. Additionally, we have performed an ablation experiment, denoted as *obj-obj*, without environmental inference by retrieving the object directly from all rooms. Table 3 shows the success rate of the object retrieval task. We demonstrate that exploring only a small number of relevant rooms is sufficient to achieve competitive performance through sub-environment recognition. In addition, it is observed that SSL models are more robust than metric learning models when the data is less structured. More details and results are shown in Appendix D.

## 7 CONCLUSION

In this paper, we propose a novel empirical method for sub-environment recognition. The fundamental relationships for sub-environment recognition, object-object, object-environment, and environment-environment relationships can be measured with metric learning models via kernel density estimation. The proposed measures successfully estimate the exact values in both openworld and photorealistic environments, and their applicability is qualitatively demonstrated. Using the OBSER framework, we claim that the agent can perform more composite inferences by chaining inferences with the proposed measures.

Our main contribution to this work is extending the use of metric learning to measure environmental relationships. Therefore, experiments have been conducted under the assumption that data acquisition for object observation is feasible. We believe that by integrating the proposed method with modules of embodied agents, such as object recognition (Kirillov et al., 2023) or navigation, the agent can perform robust and effective long-horizon inference at the level of sub-environments.

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

# A  THEORETICAL DETAILS AND PROOFS

## A.1  ADDITIONAL DEFINITIONS

### A.1.1  MEMBERSHIP FUNCTION

A membership function $I_c(x; \mu, \mathcal{F}) := p(c|x; \mathcal{F})$ with a mapping function $\mathcal{F} : \mathcal{X} \to \mathcal{Z}$ is defined as follows:

$$I_c(x; \mu, \mathcal{F}) := p(c|x; \mu, \mathcal{F}), \quad \sum_{c \in \mathcal{C}} I_c(x; \mu, \mathcal{F}) = 1.$$

By this definition, when the function $\mathcal{F}$ is optimal, the membership function works as an indicator function which does not depend on data distributions.

$$I_c^*(x) = I_c^*(x; \mu) = I_c^*(x; \nu) = \begin{cases} 1 & x \in \mathcal{X}_c \\ 0 & O.W. \end{cases} \tag{12}$$

With Definition 12 and the definition of the sub-environment, the following equation is satisfied with every function $f$.

$$\mathbb{E}_{x' \sim \mu} \left[ f(\cdot) \cdot I_c^*(x') \right] = \rho_c^{(\mu)} \cdot \mathbb{E}_{x' \sim \mathcal{D}_c} \left[ f(\cdot) \right] \tag{13}$$

### A.1.2  MESSAGE PASSING

With a kernel $\mathcal{K}_{\mathcal{Z}}(x, x'; \mathcal{F}) : \mathcal{X} \times \mathcal{X} \to [0, 1]$ defined in Equation 1, a membership function $I_c(x; \mu, \mathcal{F})$ with a data $x$ can be derived via kernel density estimation:

$$I_c(x; \mu, \mathcal{F}) = \frac{1}{Z} \cdot \mathbb{E}_{x' \sim \mu} \left[ \mathcal{K}_{\mathcal{Z}}(x, x'; \mathcal{F}) I_c^*(x') \right], \tag{14}$$

$$Z = \sum_{c \in \mathcal{C}} \mathbb{E}_{x' \sim \mu} \left[ \mathcal{K}_{\mathcal{Z}}(x, x'; \mathcal{F}) I_c^*(x') \right] = \mathbb{E}_{x' \sim \mu} \left[ \mathcal{K}_{\mathcal{Z}}(x, x'; \mathcal{F}) \right]. \tag{15}$$

$$\begin{aligned}
\therefore I_c(x; \mu, \mathcal{F}) &= \frac{\mathbb{E}_{x' \sim \mu} \left[ \mathcal{K}_{\mathcal{Z}}(x, x'; \mathcal{F}) \cdot I_c^*(x') \right]}{\mathbb{E}_{x_k \sim \mu} \left[ d(x, x_k; \mathcal{F}) \right]} \\
&= \rho_c^{(\mu)} \cdot \frac{\mathbb{E}_{x' \sim \mathcal{D}_c} \left[ \mathcal{K}_{\mathcal{Z}}(x, x'; \mathcal{F}) \right]}{\mathbb{E}_{x' \sim \mu} \left[ \mathcal{K}_{\mathcal{Z}}(x, x'; \mathcal{F}) \right]} \\
&= \frac{\rho_c^{(\mu)} \cdot \mathbb{E}_{x' \sim \mathcal{D}_c} \left[ \mathcal{K}_{\mathcal{Z}}(x, x'; \mathcal{F}) \right]}{\mathbb{E}_{x' \sim \mu} \left[ \mathcal{K}_{\mathcal{Z}}(x, x'; \mathcal{F}) \right]} \\
&= \frac{\rho_c^{(\mu)} \cdot \mathbb{E}_{x' \sim \mathcal{D}_c} \left[ \mathcal{K}_{\mathcal{Z}}(x, x'; \mathcal{F}) \right]}{\sum_{c' \in \mathcal{C}} \rho_{c'}^{(\mu)} \cdot \mathbb{E}_{x' \sim \mathcal{D}_{c'}} \left[ \mathcal{K}_{\mathcal{Z}}(x, x'; \mathcal{F}) \right]}.
\end{aligned} \tag{16}$$

## A.2  KL DIVERGENCE BETWEEN $p(c, x; \mu)$ AND $p(c, x; \mu, \mathcal{F})$

**Lemma 1** (KL divergence between $p(c, x; \mu)$ and $p(c, x; \mu, \mathcal{F})$). *Rederived form of Proposition 1 in Choi et al. (2024)*

$$D_{KL}(p(c, x; \mu) || p(c, x; \mu, \mathcal{F})) = \mathbb{E}_{x \sim \mu} \left[ -\log \frac{\mathbb{E}_{x' \sim \mathcal{D}_+} \left[ \mathcal{K}_{\mathcal{Z}}(x, x'; \mathcal{F}) \right]}{\mathbb{E}_{x' \sim \mu} \left[ \mathcal{K}_{\mathcal{Z}}(x, x'; \mathcal{F}) \right]} \right] + \mathcal{H}(\mathcal{C}; \mu). \tag{17}$$

*Proof.*

$$D_{KL}(p(c,x;\mu)||p(c,x;\mu,\mathcal{F})) \tag{18}$$

$$= \sum_{c \in \mathcal{C}} \int -p(c,x;\mu) \log \frac{p(c,x;\mu,\mathcal{F})}{p(c,x;\mu)} dx \tag{19}$$

$$= \sum_{c \in \mathcal{C}} \int -I_c^*(x) \log \frac{I_c(x;\mu,\mathcal{F})}{I_c^*(x)} d\mu(x) \tag{20}$$

$$= \sum_{c \in \mathcal{C}} \mathbb{E}_{x \sim \mu} \left[ -I_c^*(x) \log \frac{I_c(x;\mu,\mathcal{F})}{I_c^*(x)} \right] \tag{21}$$

$$= \sum_{c \in \mathcal{C}} \mathbb{E}_{x \sim \mu} \left[ -I_c^*(x) \log I_c(x;\mu,\mathcal{F}) \right] \qquad \cdots \text{ (Cross Entropy)} \tag{22}$$

$$= \sum_{c \in \mathcal{C}} \rho_c^{(\mu)} \cdot \mathbb{E}_{x \sim \mathcal{D}_c} \left[ -\log I_c(x;\mu,\mathcal{F}) \right] \tag{23}$$

By using Message Passing in Equation 16,

$$\sum_{c \in \mathcal{C}} \rho_c^{(\mu)} \cdot \mathbb{E}_{x \sim \mathcal{D}_c} \left[ -\log I_c(x;\mu,\mathcal{F}) \right] \tag{24}$$

$$= \sum_{c \in \mathcal{C}} \rho_c^{(\mu)} \cdot \mathbb{E}_{x \sim \mathcal{D}_c} \left[ -\log(\rho_c^{(\mu)} \cdot \frac{\mathbb{E}_{x' \sim \mathcal{D}_c} \left[ \mathcal{K}_{\mathcal{Z}}(x,x';\mathcal{F}) \right]}{\mathbb{E}_{x' \sim \mu} \left[ \mathcal{K}_{\mathcal{Z}}(x,x';\mathcal{F}) \right]}) \right] \tag{25}$$

$$= \sum_{c \in \mathcal{C}} \rho_c^{(\mu)} \cdot \mathbb{E}_{x \sim \mathcal{D}_c} \left[ -\log \rho_c^{(\mu)} - \log \frac{\mathbb{E}_{x' \sim \mathcal{D}_c} \left[ \mathcal{K}_{\mathcal{Z}}(x,x';\mathcal{F}) \right]}{\mathbb{E}_{x' \sim \mu} \left[ \mathcal{K}_{\mathcal{Z}}(x,x';\mathcal{F}) \right]} \right] \tag{26}$$

$$= \sum_{c \in \mathcal{C}} \rho_c^{(\mu)} \cdot \mathbb{E}_{x \sim \mathcal{D}_c} \left[ -\log \frac{\mathbb{E}_{x' \sim \mathcal{D}_c} \left[ \mathcal{K}_{\mathcal{Z}}(x,x';\mathcal{F}) \right]}{\mathbb{E}_{x' \sim \mu} \left[ \mathcal{K}_{\mathcal{Z}}(x,x';\mathcal{F}) \right]} \right] + \mathcal{H}(\mathcal{C};\mu) \tag{27}$$

$$= \mathbb{E}_{x \sim \mu} \left[ -\log \frac{\mathbb{E}_{x' \sim \mathcal{D}_+} \left[ \mathcal{K}_{\mathcal{Z}}(x,x';\mathcal{F}) \right]}{\mathbb{E}_{x' \sim \mu} \left[ \mathcal{K}_{\mathcal{Z}}(x,x';\mathcal{F}) \right]} \right] + \mathcal{H}(\mathcal{C};\mu) \tag{28}$$

$$\therefore D_{\mathrm{KL}}(p(c,x;\mu)||p(c,x;\mu,\mathcal{F})) = \mathbb{E}_{x \sim \mu} \left[ -\log \frac{\mathbb{E}_{x' \sim \mathcal{D}_+} \left[ \mathcal{K}_{\mathcal{Z}}(x,x';\mathcal{F}) \right]}{\mathbb{E}_{x' \sim \mu} \left[ \mathcal{K}_{\mathcal{Z}}(x,x';\mathcal{F}) \right]} \right] + \mathcal{H}(\mathcal{C};\mu) \tag{29}$$

$\because D_{\mathrm{KL}}(\cdot||\cdot) \geq 0,$

$$\mathbb{E}_{x \sim \mu} \left[ -\log \frac{\mathbb{E}_{x' \sim \mathcal{D}_+} \left[ \mathcal{K}_{\mathcal{Z}}(x,x';\mathcal{F}) \right]}{\mathbb{E}_{x' \sim \mu} \left[ \mathcal{K}_{\mathcal{Z}}(x,x';\mathcal{F}) \right]} \right] \geq -\mathcal{H}(\mathcal{C};\mu). \tag{30}$$

The equality is satisfied with $p(c,x;\mu) = p(c,x;\mu,\mathcal{F})$. $\qquad \square$

### A.3 OPTIMIZATION OF $(\epsilon, \delta)$ STATISTICALLY SEPARABLE (EDS) FUNCTION $\mathcal{F}_{\epsilon,\delta}$

**Theorem 1** (Optimization of $\mathcal{F}_{\epsilon,\delta}$ via generalized metric learning). *For an EDS function $\mathcal{F}_{\epsilon,\delta}$, let $\exists k \geq 1, \delta = k \cdot \epsilon$. The upperbound of $D_{KL}(p(c,x;\mu)||p(c,x;\mu,\mathcal{F}_{\epsilon,\delta}))$ is derived as:*

$$0 \leq D_{KL}(p(c,x;\mu)||p(c,x;\mu,\mathcal{F}_{\epsilon,\delta})) \leq \log\left(1 + \frac{|\mathcal{C}|-1}{k}\right) := \Delta\mathcal{H}, \tag{31}$$

*and if $\Delta\mathcal{H} \to +0$, then $k \to \infty$.*

*Proof.* With Lemma 1,

$$\sum_{c \in \mathcal{C}} \rho_c^{(\mu)} \cdot \mathbb{E}_{x \sim \mathcal{D}_c} \left[ -\log I_c(x; \mu, \mathcal{F}_{\epsilon, \delta}) \right] \tag{32}$$

$$= \mathbb{E}_{x \sim \mu} \left[ -\log \frac{\mathbb{E}_{x' \sim \mathcal{D}_+} [\mathcal{K}_{\mathcal{Z}}(x, x'; \mathcal{F}_{\epsilon, \delta})]}{\mathbb{E}_{x' \sim \mu} [\mathcal{K}_{\mathcal{Z}}(x, x'; \mathcal{F}_{\epsilon, \delta})]} \right] + \mathcal{H}(\mathcal{C}; \mu) \tag{33}$$

$$= \sum_{c \in \mathcal{C}} \rho_c^{(\mu)} \cdot \left( \mathbb{E}_{x \sim \mathcal{D}_c} \left[ -\log \mathbb{E}_{x' \sim \mathcal{D}_c} [\mathcal{K}_{\mathcal{Z}}(x, x'; \mathcal{F}_{\epsilon, \delta})] \right] + \mathbb{E}_{x \sim \mathcal{D}_c} \left[ \log \mathbb{E}_{x' \sim \mu} [\mathcal{K}_{\mathcal{Z}}(x, x'; \mathcal{F}_{\epsilon, \delta})] \right] \right) + \mathcal{H}(\mathcal{C}; \mu) \tag{34}$$

$$\leq \sum_{c \in \mathcal{C}} \rho_c^{(\mu)} \cdot \left( -\log \delta + \log(\rho_c^{(\mu)} \cdot \delta + \epsilon \cdot (1 - \rho_c^{(\mu)})) \right) + \mathcal{H}(\mathcal{C}; \mu) \tag{35}$$

$$= \sum_{c \in \mathcal{C}} \rho_c^{(\mu)} \cdot \log(1 + \frac{\epsilon \cdot (1 - \rho_c^{(\mu)})}{\delta \cdot \rho_c^{(\mu)}}) \tag{36}$$

$$\leq \log(1 + \frac{\epsilon \cdot (|\mathcal{C}| - 1)}{\delta}) = \log(1 + \frac{(|\mathcal{C}| - 1)}{k}) := \Delta \mathcal{H}. \tag{37}$$

By rewriting $k$ in terms of $\Delta \mathcal{H}$, we can get $k = \frac{|\mathcal{C}| - 1}{\exp(\Delta \mathcal{H}) - 1}$. Thus, if $\Delta \mathcal{H} \to +0$ then $k \to \infty$. $\quad\square$

### A.4 Object existence probability estimation with EDS function

**Lemma 2** (Object existence probability with EDS function). *For an EDS function $\mathcal{F}_{\epsilon, \delta}$, the proposed measure has the bound of:*

$$\hat{\rho}_{c|x}^{(\mu)} := \mathbb{E}_{x' \sim \mu} \left[ \mathcal{K}_{\mathcal{Z}}(x, x'; \mathcal{F}_{\epsilon, \delta}) \right], \quad \delta \leq \frac{\hat{\rho}_{c|x}^{(\mu)}}{\rho_c^{(\mu)} + \epsilon \cdot (1 - \rho_c^{(\mu)})} \leq 1. \tag{38}$$

*Proof.* By the definition of the EDS function,

$$\delta \cdot (\rho_c^{(\mu)} + \epsilon \cdot (1 - \rho_c^{(\mu)})) \leq \hat{\rho}_{c|x}^{(\mu)} \leq \rho_c^{(\mu)} + \epsilon \cdot (1 - \rho_c^{(\mu)}) \tag{39}$$

$$\therefore \delta \leq \frac{\hat{\rho}_{c|x}^{(\mu)}}{\rho_c^{(\mu)} + \epsilon \cdot (1 - \rho_c^{(\mu)})} \leq 1. \tag{40}$$

In the case of $\delta \to 1$, $\epsilon \to 0$, $\hat{\rho}_{c|x}^{(\mu)} \to \rho_c^{(\mu)}$. $\quad\square$

### A.5 KL divergence btw two distributions

By the definition of sub-environment, KL divergence between the distributions $\mu$ and $\nu$ of sub-environments is computed as:

$$D_{\text{KL}}(\mu || \nu) = \mathbb{E}_{x \sim \mu} \left[ \log \frac{\mu}{\nu} \right]$$

$$= \sum_{c \in \mathcal{C}} \rho_c^{(\mu)} \cdot \mathbb{E}_{x \sim \mathcal{D}_c} \left[ \log \frac{\sum_{c'} \rho_{c'}^{(\mu)} \mathcal{D}_{c'}(x)}{\sum_{c'} \rho_{c'}^{(\nu)} \mathcal{D}_{c'}(x)} \right]$$

$$= \sum_{c \in \mathcal{C}} \rho_c^{(\mu)} \cdot \mathbb{E}_{x \sim \mathcal{D}_c} \left[ \log \frac{\sum_{c'} \rho_{c'}^{(\mu)} \mathcal{D}_{c'}(x) \mathcal{D}_c(x)}{\sum_{c'} \rho_{c'}^{(\nu)} \mathcal{D}_{c'}(x) \mathcal{D}_c(x)} \right].$$

Because $\forall c' \neq c, \mathcal{D}_{c'}(x) \cdot \mathcal{D}_c(x) = 0$,

$$= \sum_{c \in \mathcal{C}} \rho_c^{(\mu)} \cdot \log \frac{\rho_c^{(\mu)}}{\rho_c^{(\nu)}}. \tag{41}$$

However, since we assume that the latent class $c$ is not directly accessible, we need an nonparametric method to approximate $D_{\text{KL}}(p(c, x; \mu, \mathcal{F})||p(c, x; \nu, \mathcal{F}))$. First we define the pseudo-divergence $D_\mu(p(c, x; \mu, \mathcal{F})||p(c, x; \nu, \mathcal{F}))$:

$$D_\mu(p(c, x; \mu, \mathcal{F})||p(c, x; \nu, \mathcal{F})) := \sum_{c \in \mathcal{C}} \left( \rho_c^{(\mu)} \cdot \mathbb{E}_{x \sim \mathcal{D}_c} \left[ \log \frac{p(c, x; \mu, \mathcal{F})}{p(c, x; \nu, \mathcal{F})} \right] \right). \tag{42}$$

By rewriting the pseudo-divergence, we obtain the kernel density based KL divergence $\hat{D}_{\text{KL}}(\mu||\nu; \mathcal{F})$ without using the latent class.

**Lemma 3** (Property of pseudo KL divergence)**.**

$$D_\mu(p(c, x; \mu, \mathcal{F})||p(c, x; \nu, \mathcal{F})) = - \underbrace{\mathbb{E}_{x \sim \mu} \left[ \log \frac{\mathbb{E}_{x' \sim \mu} [\mathcal{K}_{\mathcal{Z}}(x, x'; \mathcal{F})]}{\mathbb{E}_{x' \sim \nu} [\mathcal{K}_{\mathcal{Z}}(x, x'; \mathcal{F})]} \right]}_{\hat{D}_{KL}(\mu||\nu; \mathcal{F})} + 2 \cdot D_{KL}(\mu||\nu) \tag{43}$$

*Proof.*

$$D_\mu(p(c, x; \mu, \mathcal{F})||p(c, x; \nu, \mathcal{F})) := \sum_{c \in \mathcal{C}} \left( \rho_c^{(\mu)} \cdot \mathbb{E}_{x \sim \mathcal{D}_c} \left[ \log \frac{p(c, x; \mu, \mathcal{F})}{p(c, x; \nu, \mathcal{F})} \right] \right) \tag{44}$$

$$= D_{\text{KL}}(p(c, x; \mu)||p(c, x; \nu, \mathcal{F})) - D_{\text{KL}}(p(c, x; \mu)||p(c, x; \mu, \mathcal{F})) \tag{45}$$

i) $D_{\text{KL}}(p(c, x; \mu)||p(c, x; \nu, \mathcal{F}))$

$$D_{\text{KL}}(p(c, x; \mu)||p(c, x; \nu, \mathcal{F})) \tag{46}$$

By utilizing the same method as in Lemma 1,

$$= \sum_{c \in \mathcal{C}} \rho_c^{(\mu)} \cdot \mathbb{E}_{x \sim \mathcal{D}_c} \left[ - \log I_c(x; \nu, \mathcal{F}) \right] + D_{\text{KL}}(\mu||\nu) \tag{47}$$

$$= \sum_{c \in \mathcal{C}} \rho_c^{(\mu)} \cdot \mathbb{E}_{x \sim \mathcal{D}_c} \left[ - \log \rho_c^{(\nu)} - \log \frac{\mathbb{E}_{x' \sim \mathcal{D}_c} [\mathcal{K}_{\mathcal{Z}}(x, x'; \mathcal{F})]}{\mathbb{E}_{x' \sim \nu} [\mathcal{K}_{\mathcal{Z}}(x, x'; \mathcal{F})]} \right] + D_{\text{KL}}(\mu||\nu) \tag{48}$$

$$= \mathbb{E}_{x \sim \mu} \left[ - \log \frac{\mathbb{E}_{x' \sim \mathcal{D}_+} [\mathcal{K}_{\mathcal{Z}}(x, x'; \mathcal{F})]}{\mathbb{E}_{x' \sim \nu} [\mathcal{K}_{\mathcal{Z}}(x, x'; \mathcal{F})]} \right] + D_{\text{KL}}(\mu||\nu) + \text{CE}(\mu||\nu) \tag{49}$$

ii) $D_{\text{KL}}(p(c, x; \mu)||p(c, x; \mu, \mathcal{F}))$ (Lemma 1)

$$D_{\text{KL}}(p(c, x; \mu)||p(c, x; \mu, \mathcal{F})) = \mathbb{E}_{x \sim \mu} \left[ - \log \frac{\mathbb{E}_{x' \sim \mathcal{D}_+} [\mathcal{K}_{\mathcal{Z}}(x, x'; \mathcal{F})]}{\mathbb{E}_{x' \sim \mu} [\mathcal{K}_{\mathcal{Z}}(x, x'; \mathcal{F})]} \right] + \mathcal{H}(C; \mu) \tag{50}$$

Putting together,

$$\therefore D_\mu(p(c, x; \mu, \mathcal{F})||p(c, x; \nu, \mathcal{F})) \tag{51}$$

$$\begin{aligned} &= \mathbb{E}_{x \sim \mu} \left[ - \log \frac{\mathbb{E}_{x' \sim \mathcal{D}_+} [\mathcal{K}_{\mathcal{Z}}(x, x'; \mathcal{F})]}{\mathbb{E}_{x' \sim \nu} [\mathcal{K}_{\mathcal{Z}}(x, x'; \mathcal{F})]} \right] + D_{\text{KL}}(\mu||\nu) + \text{CE}(\mu||\nu) \\ &\quad - \mathbb{E}_{x \sim \mu} \left[ - \log \frac{\mathbb{E}_{x' \sim \mathcal{D}_+} [\mathcal{K}_{\mathcal{Z}}(x, x'; \mathcal{F})]}{\mathbb{E}_{x' \sim \mu} [\mathcal{K}_{\mathcal{Z}}(x, x'; \mathcal{F})]} \right] - \mathcal{H}(C; \mu) \end{aligned} \tag{52}$$

$$= 2 \cdot D_{\text{KL}}(\mu||\nu) - \hat{D}_{\text{KL}}(\mu||\nu; \mathcal{F}). \tag{53}$$

$\square$

Intuitively, when $\mathcal{F} \to \mathcal{F}^*$ with $p(c, x; \mu) = p(c, x; \mu, \mathcal{F}^*)$, then $D_\mu(p(c, x; \mu, \mathcal{F})||p(c, x; \nu, \mathcal{F})) \to D_{\text{KL}}(\mu||\nu)$ is satisfied. Therefore, we can suppose $\hat{D}_{\text{KL}}(\mu||\nu; \mathcal{F}) \to D_{\text{KL}}(\mu||\nu)$ when $\mathcal{F} \to \mathcal{F}^*$.

## A.6 Approximation on KL divergence with EDS function

**Theorem 2** (KL divergence with $\mathcal{F}_{\epsilon,\delta}$). *For an EDS function $\mathcal{F}_{\epsilon,\delta}$, the proposed measure in Definition 3 has the bound of:*

$$\left| \hat{D}_{KL}(\mu||\nu; \mathcal{F}_{\epsilon,\delta}) - \sum_{c \in \mathcal{C}} \rho_c^{(\mu)} \cdot \log \left( \frac{\rho_c^{(\mu)} + (1 - \rho_c^{(\mu)}) \cdot \epsilon}{\rho_c^{(\nu)} + (1 - \rho_c^{(\nu)}) \cdot \epsilon} \right) \right| \leq -\log \delta, \tag{54}$$

*in $(\mu, \nu)$-almost everywhere. With optimized $\mathcal{F}_{\epsilon,\delta}$, such that $\delta \to 1, \epsilon \to 0$, $\hat{D}_{KL}(\mu||\nu; \mathcal{F}_{\epsilon,\delta})$ converges to $D_{KL}(\mu||\nu)$.*

*Proof.*

$$\mathbb{E}_{x \sim \mu} \left[ \log \frac{\mathbb{E}_{x' \sim \mu} [\mathcal{K}_{\mathcal{Z}}(x, x'; \mathcal{F}_{\epsilon,\delta})]}{\mathbb{E}_{x' \sim \nu} [\mathcal{K}_{\mathcal{Z}}(x, x'; \mathcal{F}_{\epsilon,\delta})]} \right] \tag{55}$$

$$= \sum_{c \in \mathcal{C}} \rho_c^{(\mu)} \cdot \left( \mathbb{E}_{x \sim \mathcal{D}_c} \left[ \log \frac{\sum_{c' \in \mathcal{C}} \rho_{c'}^{(\mu)} \cdot \mathbb{E}_{x' \sim \mathcal{D}_{c'}} [\mathcal{K}_{\mathcal{Z}}(x, x'; \mathcal{F}_{\epsilon,\delta})]}{\sum_{c' \in \mathcal{C}} \rho_{c'}^{(\nu)} \cdot \mathbb{E}_{x' \sim \mathcal{D}_{c'}} [\mathcal{K}_{\mathcal{Z}}(x, x'; \mathcal{F}_{\epsilon,\delta})]} \right] \right) \tag{56}$$

$$\leq \sum_{c \in \mathcal{C}} \rho_c^{(\mu)} \cdot \log \left( \mathbb{E}_{x \sim \mathcal{D}_c} \left[ \frac{\sum_{c' \in \mathcal{C}} \rho_{c'}^{(\mu)} \cdot \mathbb{E}_{x' \sim \mathcal{D}_{c'}} [\mathcal{K}_{\mathcal{Z}}(x, x'; \mathcal{F}_{\epsilon,\delta})]}{\sum_{c' \in \mathcal{C}} \rho_{c'}^{(\nu)} \cdot \mathbb{E}_{x' \sim \mathcal{D}_{c'}} [\mathcal{K}_{\mathcal{Z}}(x, x'; \mathcal{F}_{\epsilon,\delta})]} \right] \right) \tag{57}$$

$$\leq \sum_{c \in \mathcal{C}} \rho_c^{(\mu)} \cdot \log \left( \frac{\rho_c^{(\mu)} + (1 - \rho_c^{(\mu)}) \cdot \epsilon}{\rho_c^{(\nu)} + (1 - \rho_c^{(\nu)}) \cdot \epsilon} \right) - \log \delta. \tag{58}$$

$$\tag{59}$$

In a same way,

$$\mathbb{E}_{x \sim \mu} \left[ \log \frac{\mathbb{E}_{x' \sim \mu} [\mathcal{K}_{\mathcal{Z}}(x, x'; \mathcal{F}_{\epsilon,\delta})]}{\mathbb{E}_{x' \sim \nu} [\mathcal{K}_{\mathcal{Z}}(x, x'; \mathcal{F}_{\epsilon,\delta})]} \right] \tag{60}$$

$$= -\sum_{c \in \mathcal{C}} \rho_c^{(\mu)} \cdot \left( \mathbb{E}_{x \sim \mathcal{D}_c} \left[ \log \frac{\sum_{c' \in \mathcal{C}} \rho_{c'}^{(\nu)} \cdot \mathbb{E}_{x' \sim \mathcal{D}_{c'}} [\mathcal{K}_{\mathcal{Z}}(x, x'; \mathcal{F}_{\epsilon,\delta})]}{\sum_{c' \in \mathcal{C}} \rho_{c'}^{(\mu)} \cdot \mathbb{E}_{x' \sim \mathcal{D}_{c'}} [\mathcal{K}_{\mathcal{Z}}(x, x'; \mathcal{F}_{\epsilon,\delta})]} \right] \right) \tag{61}$$

$$\geq -\sum_{c \in \mathcal{C}} \rho_c^{(\mu)} \cdot \log \left( \mathbb{E}_{x \sim \mathcal{D}_c} \left[ \frac{\sum_{c' \in \mathcal{C}} \rho_{c'}^{(\nu)} \cdot \mathbb{E}_{x' \sim \mathcal{D}_{c'}} [\mathcal{K}_{\mathcal{Z}}(x, x'; \mathcal{F}_{\epsilon,\delta})]}{\sum_{c' \in \mathcal{C}} \rho_{c'}^{(\mu)} \cdot \mathbb{E}_{x' \sim \mathcal{D}_{c'}} [\mathcal{K}_{\mathcal{Z}}(x, x'; \mathcal{F}_{\epsilon,\delta})]} \right] \right) \tag{62}$$

$$\geq -\sum_{c \in \mathcal{C}} \rho_c^{(\mu)} \cdot \log \left( \frac{\rho_c^{(\nu)} + (1 - \rho_c^{(\nu)}) \cdot \epsilon}{\rho_c^{(\mu)} + (1 - \rho_c^{(\mu)}) \cdot \epsilon} \right) + \log \delta \tag{63}$$

$$\geq \sum_{c \in \mathcal{C}} \rho_c^{(\mu)} \cdot \log \left( \frac{\rho_c^{(\mu)} + (1 - \rho_c^{(\mu)}) \cdot \epsilon}{\rho_c^{(\nu)} + (1 - \rho_c^{(\nu)}) \cdot \epsilon} \right) + \log \delta. \tag{64}$$

$\square$

Note that when $\delta \to 1$ and $\epsilon \to 0$, the measure $\hat{D}_{KL}(\mu||\nu; \mathcal{F}_{\epsilon,\delta})$ converges to $D_{KL}(\mu||\nu)$.

# B Experiment Details

## B.1 Toy Problem

We first validate the EDS function and its behavior during the optimization with several elementary environments: Moons and XOR datasets. We choose the hypersphere as the embedding space. A shallow MLP structure is chosen as the feature extractor for both experiments: the number of nodes for each layer is selected as (2,8,4,2), respectively. The feature extractor is trained with a balanced set in 30 epochs. With the trained representations, we measure $\epsilon$ and $\delta$, and show that the proposed approximated measures are accurate with an optimized EDS function.

## B.2 IMAGENET DATASET

For the ImageNet dataset, we evaluate several self-supervised learning models using various metrics. We use MoCo-v2 Chen et al. (2020b), MoCo-v3 Chen et al. (2021), SimCLR-v1 Chen et al. (2020a), and SupCon Khosla et al. (2020) by utilizing L2-normalized projection head features. However, for DINO (v1 Caron et al. (2021), v2 Oquab et al. (2023)), we use L2-normalized backbone features rather than projection features because the origin papers employed normalized backbone features to measure KNN accuracy. We utilize pre-trained weights which are trained with ImageNet dataset for both the backbone and projection head of each model. The reported Top-1 linear probing accuracy is sourced directly from the respective papers or GitHub repos.

For augmentation set used for evaluation, we apply the most basic augmentations: (1) resizing to $256 \times 256$, (2) center cropping to $224 \times 224$, and (3) normalizing with mean and standard deviation of ImageNet dataset. We omit normalization for experiments with SimCLR since the original implementation did not use normalization.

## B.3 MINECRAFT DATASET

In this paper, we construct a Minecraft dataset containing 26,000 images and the corresponding labels. We gather ego-centric observations of objects to train or fine-tune models. To build the dataset, we first choose typical biomes that can represent all objects in the *overworld* in Minecraft. The dataset is derived from two environments: an open-world environment and a miniature environment.

### B.3.1 OPEN-WORLD MULTI-ANGLED DATA COLLECTION

For the open-world multi-angled data collection, we use the Minecraft's default world generation settings to generate a world environment where the agent collects the dataset. In such an environment, the agents can easily get stuck due to composite terrains and the arrangement of objects. This reduces the efficiency of data acquisition. Therefore, we manually locate 100 objects per biome in the world using a fixed seed, rather than relying on fully automatic methods such as the random walk algorithm. For each object, we gather 30 observations by rotating around the object's position. Figure 11 shows examples of the observations collected. We split 10% of each observation to build the test set. Note that we do not split the dataset by object because object frequencies follow a long-tailed distribution, and object-based splitting could introduce further distortions in the data distribution. We choose two levels of hierarchical concepts to conduct experiments with different levels of abstraction. Table 4 shows the proportion of hierarchical concepts of the proposed dataset.

### B.3.2 MINIATURE ENVIRONMENT MULTI-ANGLED DATA COLLECTION

We also collected data from our miniature environment, which consists of $4 \times 5$ grids of $48 \times 48$ blocks with the same biome. The kind of 10 biomes in the dataset is the same as the ones in the open-world multi-angled data collection and in the table 4. To collect the data in the miniature environment, we used the same algorithm as used in open-world environment.

### B.3.3 MINIATURE SCENARIO COLLECTION

In the miniature scenario, the agent performs random exploration within the grid to collect visual observations programmatically. For each grid, we teleport the agent to its center. The agent wanders randomly through the grid to collect visual observations and save them as images. The agent is prevented from going outside the grid by checking its distance from the grid's center. If the agent goes farther than 20 blocks from the center of the grid, we force the agent to look at the center of the grid using the teleport command. To reduce the likelihood of the agent's view capturing parts of neighboring grids with different biomes, we have the agent look slightly downward following a normal distribution, $\mathcal{N}(30, 5^2)$. In this setting, we also use Pareto distribution to determine the distance it moves forward before turning around by degrees randomly following $\mathcal{N}(30, 1^2)$.

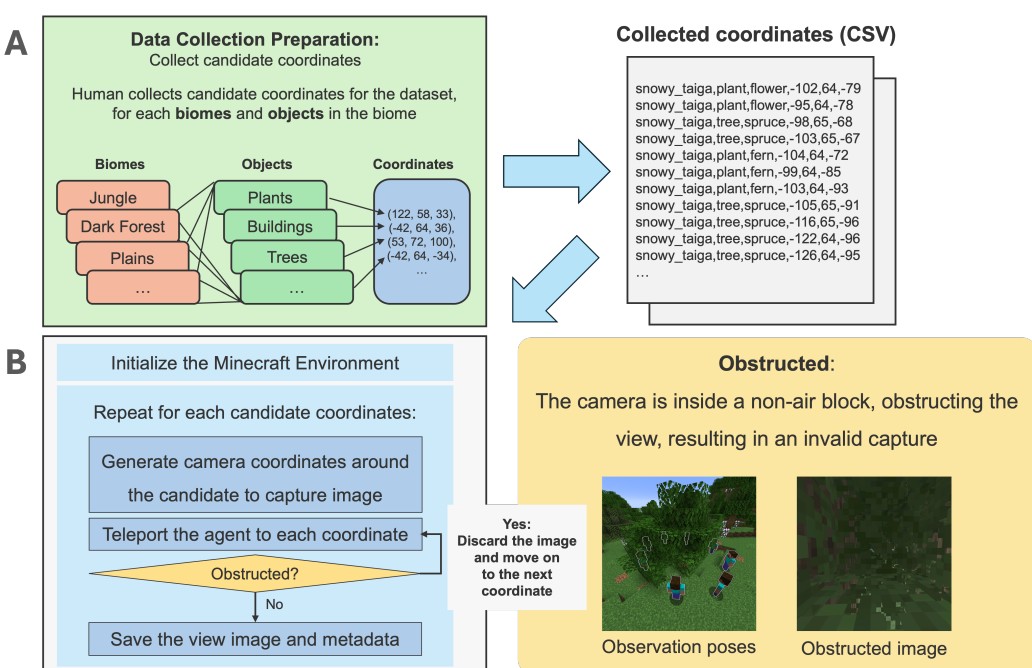

Figure 10: Flowchart illustrating the Minecraft data collection process. A) A human manually collects candidate coordinates for the dataset, with separate CSV files generated for each biome. With 10 different biomes, we gather coordinates of each biome with 10 distinct CSV files. Each CSV file contains entries for the biome name, object type, object name, and the corresponding x, y, z coordinates for each object within that biome. B) Using a customized Minecraft automation setup, the agent collects the data with each CSV files. The agent is teleported to specified coordinates and performs a 360-degree rotation around the point of interest, capturing images at 30 intervals of 12 degrees each. If the view of the camera is obstructed by being inside a non-air block, the corresponding image is automatically discarded. This process is repeated across all biomes. Successfully captured images are saved with the associated coordinates in the filename.

Table 4: Summary of statistical information for the Minecraft dataset. We focus on gathering unique objects from each biome, and it draws some difference between gathered data and real distribution, especially with villages. For evaluation, we use observations from the Miniature environment instead of gathered dataset.

| Biome | Category | Sub-category | Frequency |
|---|---|---|---|
| forest | plant | flower | 0.09 |
| | tree | oak | 0.52 |
| | | birch | 0.39 |
| dark forest | plant | flower | 0.02 |
| | | big mushroom (brown) | 0.12 |
| | | big mushroom (red) | 0.17 |
| | | pumpkin | 0.01 |
| | tree | oak | 0.16 |
| | | birch | 0.12 |
| | | dark oak | 0.40 |
| desert | plant | cactus | 0.35 |
| | | sugarcane | 0.19 |
| | | dead bush | 0.20 |
| | tree | azalea | 0.05 |
| | village | building | 0.08 |
| | | decorative | 0.08 |
| | | farm | 0.05 |
| savanna | plant | flower | 0.07 |
| | | grass | 0.20 |
| | | pumpkin | 0.05 |
| | | melon | 0.05 |
| | tree | oak | 0.09 |
| | | acacia | 0.27 |
| | village | building | 0.20 |
| | | decorative | 0.06 |
| | | farm | 0.05 |
| swamp | plant | flower | 0.09 |
| | | sugarcane | 0.14 |
| | | lily pad | 0.11 |
| | | grass | 0.06 |
| | | small mushroom (brown) | 0.16 |
| | | small mushroom (red) | 0.01 |
| | | pumpkin | 0.02 |
| | | dead bush | 0.08 |
| | tree | oak | 0.31 |
| | structure | building | 0.02 |

| Biome | Category | Sub-category | Frequency |
|---|---|---|---|
| plains | plant | flower | 0.26 |
| | | grass | 0.33 |
| | | pumpkin | 0.03 |
| | tree | oak | 0.21 |
| | village | building | 0.12 |
| | | decorative | 0.02 |
| | | farm | 0.03 |
| snowy plains | plant | flower | 0.16 |
| | | grass | 0.28 |
| | | pumpkin | 0.01 |
| | tree | spruce | 0.25 |
| | village | building | 0.21 |
| | | decorative | 0.07 |
| | | farm | 0.02 |
| taiga | plant | flower | 0.04 |
| | | fern | 0.12 |
| | | berry bush | 0.12 |
| | | pumpkin | 0.03 |
| | | small mushroom (brown) | 0.04 |
| | tree | spruce | 0.42 |
| | village | building | 0.16 |
| | | decorative | 0.04 |
| | | farm | 0.03 |
| snowy taiga | plant | flower | 0.05 |
| | | fern | 0.23 |
| | | berry bush | 0.02 |
| | | pumpkin | 0.01 |
| | | small mushroom (brown) | 0.05 |
| | tree | spruce | 0.64 |
| jungle | plant | flower | 0.02 |
| | | bamboo | 0.21 |
| | | melon | 0.11 |
| | tree | oak | 0.11 |
| | | jungle | 0.55 |

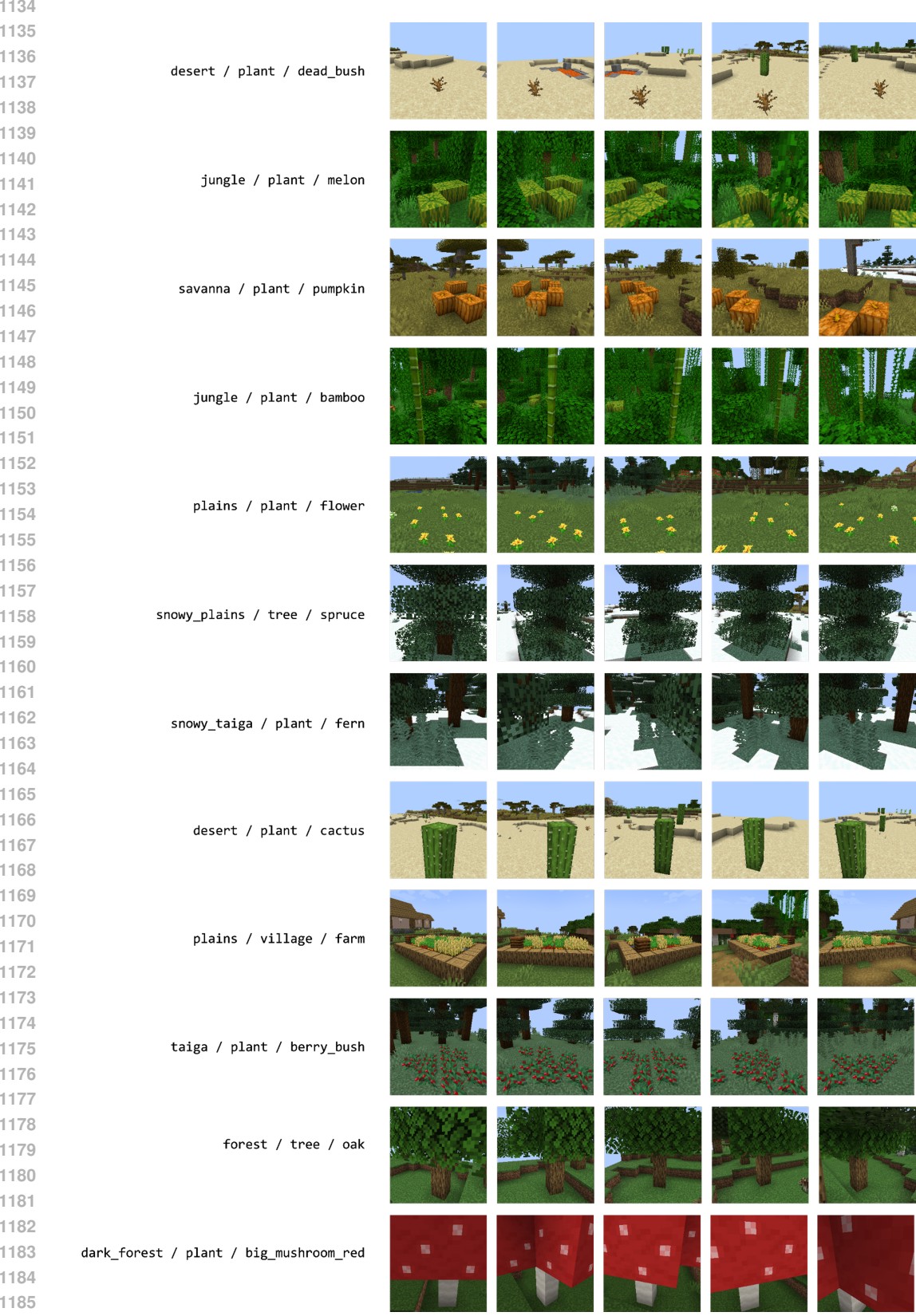

Figure 11: Example observations with various objects from different biomes.

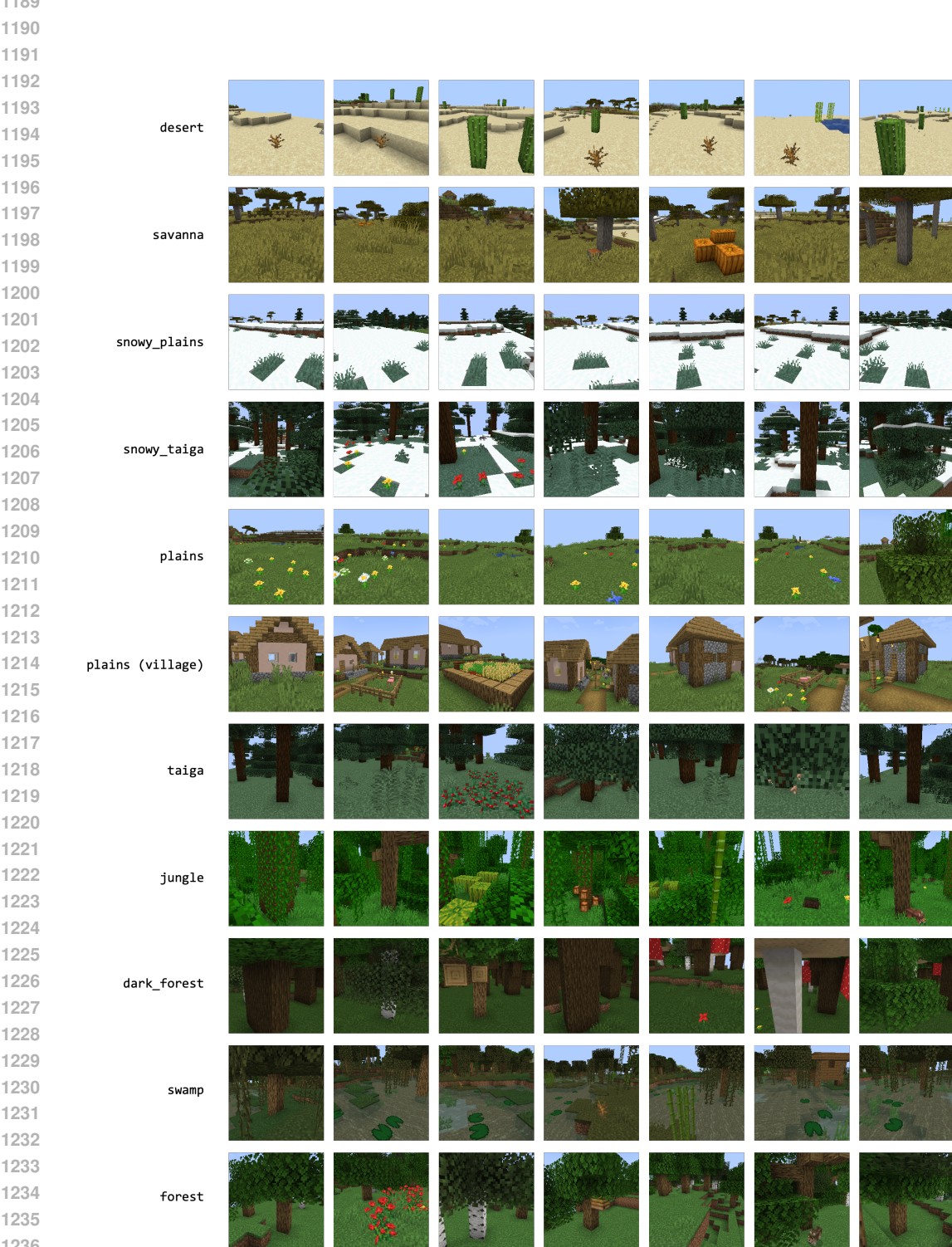

Figure 12: Example observations from different biomes.

## B.4 MINECRAFT EXPERIMENT

For experiments in Minecraft environment, we choose SimCLR, MoCo, and SupCon with ResNet-50 as a backbone. In this experiment, we utilize the projection layer of each model in the same way as it was applied to the ImageNet data in the original works. We have used 1 NVIDIA RTX 3090 Ti to train each model. Table 5 shows the hyperparameters and details used to train each model. We add `RandomResizedCrop` to the augmentation set to reflect the various distances between the agent and the object. PyTorch-style pseudocode of the augmentation set is shown in Algorithm 1.

Table 5: Details of the hyperparameters and settings used to train each model.

| Model | Backbone | Emb. Dim | Emb. Type | Batch size | Epochs (Warmup / Train) | Optimizer | Learning rate | Scheduler | Temperature |
|---|---|---|---|---|---|---|---|---|---|
| SupCon | ResNet-50 | 128 | Projection | 128 | 10 / 20 | LARS | 0.15 | Cosine | 0.2 |
| SimCLR | ResNet-50 | 128 | Projection | 128 | 10 / 20 | LARS | 0.15 | Cosine | 0.2 |
| MoCo-v2 | ResNet-50 | 128 | Projection | 128 | 10 / 100 | SGD | 0.03 | Cosine | 0.2 |

---

**Algorithm 1** PyTorch-style code of the augmentation set for Minecraft experiments.

```
transform = Compose([
    Resize(256),
    RandomResizedCrop(size=224,scale=[0.5,1.0]),
    RandomApply(
        [ColorJitter(0.2,0.2,0.2,0.1)], p=0.8
    ),
    RandomHorizontalFlip(),
    ToTensor(),
    Normalize(
        mean=[0.3232, 0.3674, 0.2973],
        std=[0.2615, 0.2647, 0.3390]
    ),
])
```

---

## B.5 ALGORITHMS OF THE PROPOSED MEASURES

---

**Algorithm 2** PyTorch-style pseudocode of existence probability estimation.

```
def existence_estimation(query,mu,tau,type_='cosine',multiplier=0.25):

    if metric == 'cosine':
        mean = fisher_rao_mean(query)
    else:
        mean = query.mean(0,keepdim=True)
    mean_mu_matrix = dist_matrix(mu,mean,metric)
    mean_mu_density = kernel(mean_mu_matrix,tau,metric).mean(-1)
    mean_query_matrix = dist_matrix(query,mean,metric)
    tol = kernel(mean_query_matrix,tau,metric).mean() * multiplier
    mean_mu_density = (mean_mu_kernel > tol).double()
    return mean_mu_density.mean(0)
```

---

**Algorithm 3** PyTorch-style pseudocode of KL divergence estimation.

```
def kldiv_estimation(mu,nu,tau,metric='cosine'):
    mu_mu_matrix = dist_matrix(mu,mu,metric)
    mu_nu_matrix = dist_matrix(mu,nu,metric)
    mu_mu_log_density = kernel(mu_mu_matrix,tau,metric).mean(-1).log()
    mu_nu_log_density = kernel(mu_nu_matrix,tau,metric).mean(-1).log()
    return (mu_mu_log_density-mu_nu_log_density).mean()
```

Table 6: Validation with toy datasets. (3 times, mean values are reported) $S_1$: $[0.5, 0.5] \rightarrow [0.1, 0.9]$, $S_2$: $[0.2, 0.8] \rightarrow [0.8, 0.2]$

| Dataset | Model | $\epsilon$ ($\downarrow$) | $\delta$ ($\uparrow$) | $\log(\delta/\epsilon)$ | $\hat{D}_{\mathrm{KL}}(S_1)$ | $\|\Delta D_{\mathrm{KL}}(S_1)\|$ | $\hat{D}_{\mathrm{KL}}(S_2)$ | $\|\Delta D_{\mathrm{KL}}(S_2)\|$ |
|---------|-------|------|------|------|------|------|------|------|
| Moons | Norm | 0.0005 | 0.0975 | 5.3098 | 0.4840 | 0.0268 | 0.8152 | 0.0166 |
| | EDS-e | 0.0000 | 0.6174 | **8.8720** | 0.5080 | 0.0030 | 0.8330 | 0.0046 |
| | EDS-s | 0.0011 | **0.9922** | 6.8669 | 0.5034 | 0.0075 | 0.8267 | 0.0051 |
| XOR | Norm | $\sim 0$ | 0.2136 | 27.0365 | 0.5163 | 0.0030 | 0.8295 | 0.0029 |
| | EDS-e | $\sim 0$ | 0.6810 | **53.7363** | 0.5111 | 0.0039 | 0.8271 | 0.0046 |
| | EDS-s | $\sim 0$ | **0.9991** | 28.5699 | **0.5108** | **0.0000** | **0.8308** | **0.0000** |

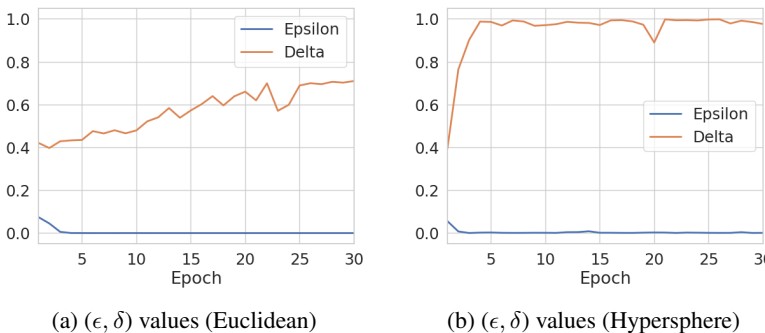

(a) $(\epsilon, \delta)$ values (Euclidean)    (b) $(\epsilon, \delta)$ values (Hypersphere)

Figure 13: Optimization of EDS functions in Euclidean and Hypersphere spaces. $\delta$ cannot reach to 1 because Euclidean space is not a compact space.

## C ADDITIONAL RESULTS

### C.1 VISUALIZATION OF TOY EXAMPLE

**Ablation. the role of the compact space**   In this work, we use hypersphere which is a compact space rather than Euclidean space. Note that optimizing the EDS function in Euclidean space, which has no upper bound on the distance, does not guarantee the improvement of the concentration property. Therefore, we claim that the embedding space should be compact to guarantee the convergence of both $\epsilon$ and $\delta$.

**Lemma 4** (Optimization of $\mathcal{F}_{\epsilon,\delta}$ in compact space). *Let* $d_{\mathcal{Z}} : \mathcal{Z} \times \mathcal{Z} \rightarrow [0, d_{\max}]$, $h_\tau(t) := \exp(-t/\tau)$. $\exists k \leq k_{\max} := \exp(d_{\max}/\tau)$, $\delta = k\epsilon$. *By Theorem 1, without any additional restrictions of* $\mathcal{Z}$, $\Delta\mathcal{H} \rightarrow \Delta\mathcal{H}_{\min}, k \rightarrow k_{\max}$.

$$\delta \rightarrow 1, \ \epsilon \rightarrow \exp(-d_{\max}/\tau). \tag{65}$$

*Proof.* Since the metric is bounded with $[0, d_{\max}]$, $\delta$ and $\epsilon$ also bounded to $[\exp(-d_{\max}/\tau), 1]$. Thus, $\delta/\epsilon = k \in [1, k_{\max}]$ with $\delta \geq \epsilon$. By Theorem 1, we can get:

$$D_{\mathrm{KL}}(p(c, x; \mu)\|p(c, x; \mu, \mathcal{F}_{\epsilon,\delta})) \leq \log\left(1 + \frac{|\mathcal{C}| - 1}{k}\right) := \Delta\mathcal{H}. \tag{66}$$

$$\therefore \exists \Delta\mathcal{H}_{\min} = \log(1 + (|\mathcal{C}| - 1)/k_{\max}) \leq \Delta\mathcal{H}, \tag{67}$$

$$k \rightarrow k_{\max}, \ \Delta\mathcal{H} \rightarrow \Delta\mathcal{H}_{\min}, \quad \delta \rightarrow 1, \ \epsilon \rightarrow \exp(-d_{\max}/\tau). \tag{68}$$

$\square$

Note that $\exp(-d_{\max}/\tau) \simeq 0$ when $\tau$ is set sufficiently small. To illustrate the difference between Euclidean space and hypersphere, we conduct toy experiments using two small datasets: Moons and XOR. For each dataset, we train models in both Euclidean and hypersphere embedding spaces, then compute the $\epsilon$ and $\delta$ values with a temperature of $\tau = 0.07$. The results are shown in Table 6. In Euclidean space, the metric between data points is unbounded, hindering the convergence of $\delta$ to 1. In contrast, on the hypersphere, $\delta$ converges to nearly 1, while $\log(k)$ remains smaller than 1 in

Euclidean space. This convergence of $\delta$ also enhances the precision of KL divergence. These results justify the use of the hypersphere as the embedding space for our experiments.

## C.2 IMAGENET EXPERIMENT

### C.2.1 EDS VALUES OF PRETRAINED MODELS

We first compute the $\epsilon$ and $\delta$ values of $\mathcal{F}_{\epsilon,\delta}$ for each trained model. To reduce the influence of outliers, we removed approximately 5% of the data before aggregating the kernel density. Additionally, we report the mean values of $\epsilon$ and $\delta$ across all classes, rather than the minimum or maximum values, as the size of each class cluster can vary. Figure 14 visualizes the EDS values for each model with different temperature settings of $\tau$.

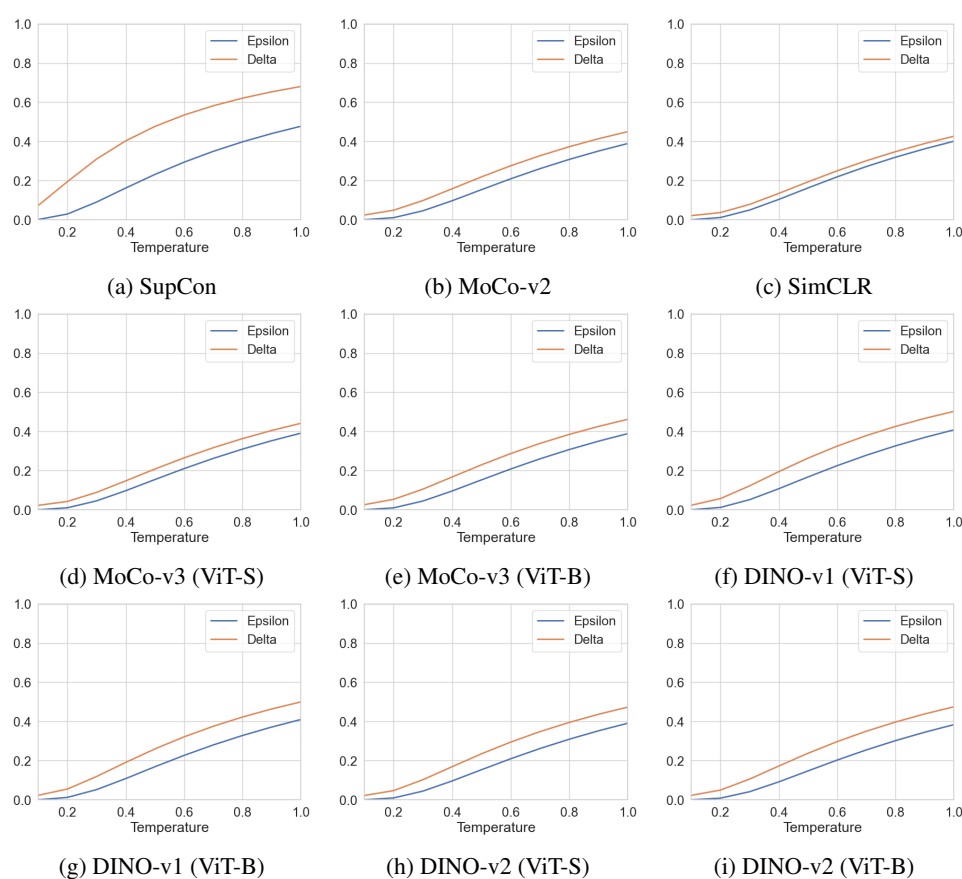

Figure 14: Visualization of EDS values with different temperatures with ImageNet Dataset.

### C.2.2 EXISTENCE PROBABILITY ESTIMATION

We conduct additional experiments with different scenarios. Figure 15 shows the results with (a) a uniform distribution and (b) a Zipf distribution with $\alpha = 0.7$. In both cases, our method successfully estimates the original distribution.

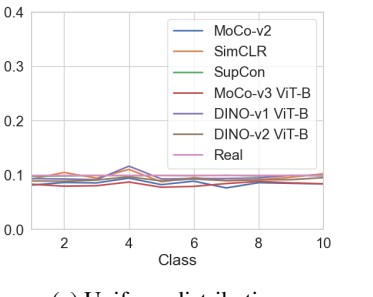 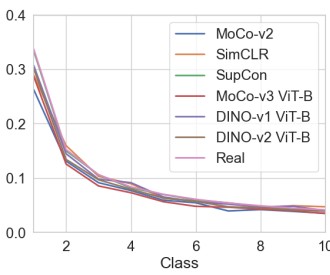

(a) Uniform distribution      (b) A Zipf distribution ($\alpha = 0.7$)

Figure 15: Visualization of object existence probability estimation with various scenarios. Mean classifier used to enhance the estimation accuracy with temperature $\tau = 0.2$ (Mean values are reported with 5 different seeds.)

### C.2.3 KL DIVERGENCE ESTIMATION

For the KL divergence estimation experiment, we take the following steps to obtain the artificial data distribution. First, we randomly select a certain number of classes and divide them into two groups. Then, we sample the data according to the proportion of each group. At this point, the optimal KL divergence is the KL divergence value corresponding to the group ratios. The KL divergence estimate is then be computed using Algorithm 3. Table 7 describes the three scenarios in which the experiment was conducted.

Table 7: Descriptions of scenarios in KL divergence experiments.

|  | Number of classes | $\mu$ | $\nu$ | Number of images | Optimal KL div. |
|---|---|---|---|---|---|
| Scenario 1 | 10 (5/5) | $[0.4, 0.6]$ | $[0.6, 0.4]$ | 1000 | 0.0811 |
| Scenario 2 | 10 (5/5) | $[0.2, 0.8]$ | $[0.8, 0.2]$ | 1000 | 0.8318 |
| Scenario 3 | 40 (20/20) | $[0.2, 0.8]$ | $[0.8, 0.2]$ | 4000 | 0.8318 |

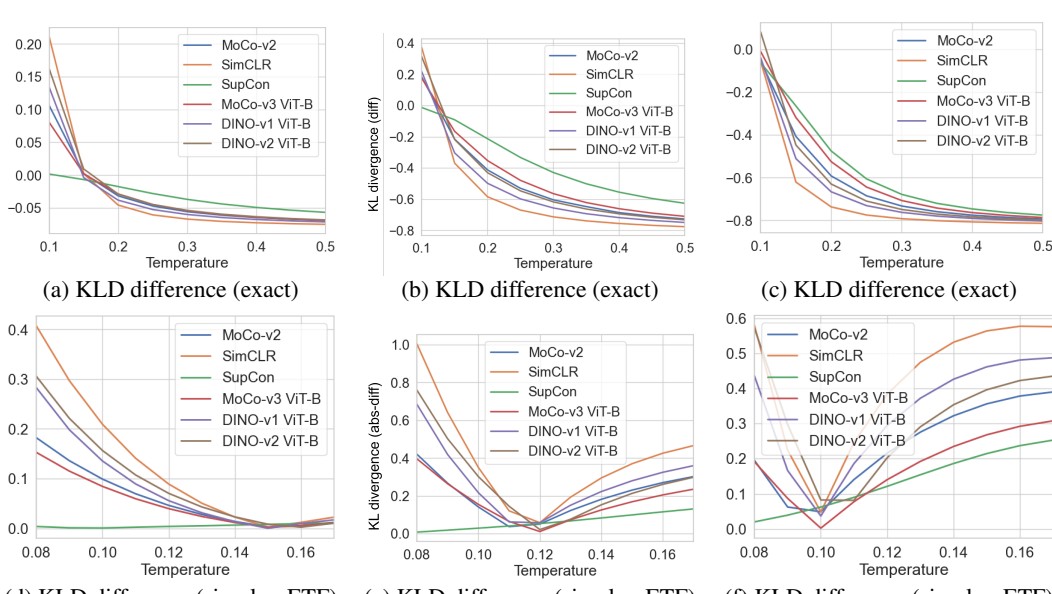

(a) KLD difference (exact)  (b) KLD difference (exact)  (c) KLD difference (exact)

(d) KLD difference (simplex-ETF)  (e) KLD difference (simplex-ETF)  (f) KLD difference (simplex-ETF)

Figure 16: Visualization of KL divergence estimation with three different scenarios. (a)-(d) correspond to Scenario 1, (b)-(e) to Scenario 2, and (c)-(f) to Scenario 3. The scenario setting is shown in Table 7. (Mean values are reported with 5 different seeds.)

### C.3 MINECRAFT EXPERIMENT

#### C.3.1 EDS VALUES OF TRAINED MODELS

As the same method in Appendix C.2, we also visualize the $\epsilon$ and $\delta$ values of each trained model in Figure 17. Due to the simpler task than ImageNet, each model has higher $\delta$ and lower $\epsilon$ than those in ImageNet.

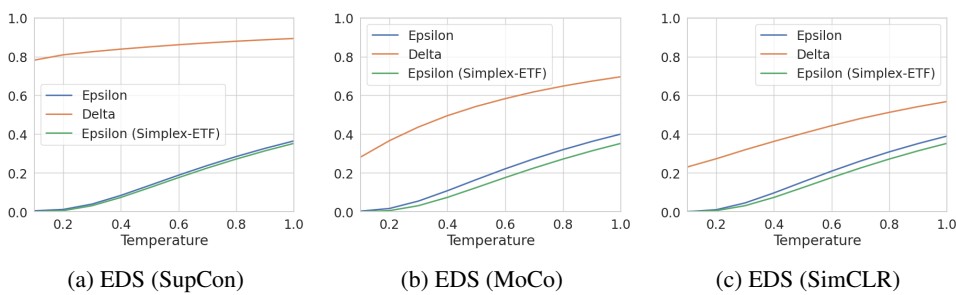

(a) EDS (SupCon)  (b) EDS (MoCo)  (c) EDS (SimCLR)

Figure 17: Visualization of EDS values with different temperatures with Minecraft Dataset.

### C.3.2 OBJECT-ENVIRONMENT RETRIEVAL TASK

The verification of the object-environment retrieval task is performed through the following process. First, observations of objects within each grid of the Miniature environment are obtained. Then, using 5 observations of the objects as a query, the existence probability is estimated for each model, followed by the visualization of heatmaps of the estimated probabilities of each models. Figure 18 shows additional results with different queries.

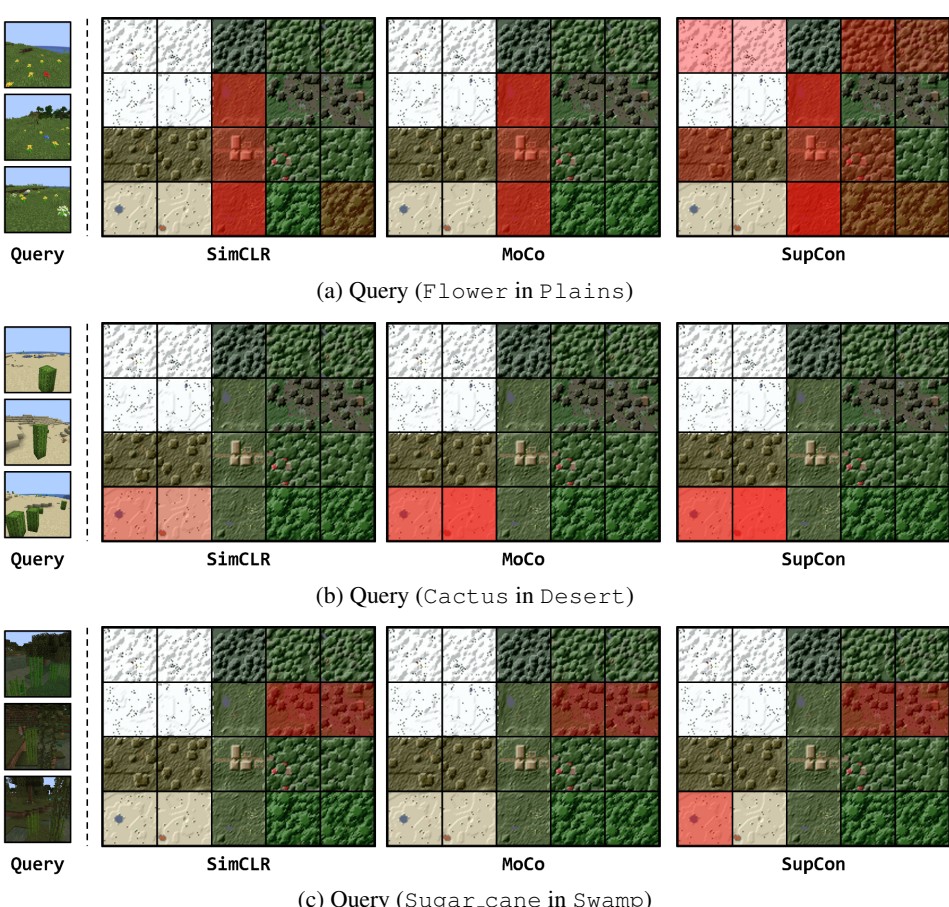

Figure 18: Visualization of results with object-environment retrieval task in Miniature environment.

### C.4 Difference between metric learning and SSL in OBSER

We visualize the pair-wise environmental relationship with the Jensen-Shannon divergence. Figure 19 shows the MDS visualization (Buja et al., 2008) of biomes in the Miniature environment with each model. The SSL models tend to keep the metrics of sub-environments farther apart, as they consider ambient biases beyond object distribution within each sub-environment. In contrast, the metric learning model better captures differences in object distribution but struggles to incorporate less task-relevant information.

We have observed that both SSL and metric learning models demonstrate the capability of making sufficiently accurate inferences for each recognition task in OBSER. We have also observed distinct differences between these paradigms. SSL models tend to integrate environmental information that is not directly relevant to the task, while metric learning models focus on task-specific information for more accurate inference. We claim that SSL and metric learning models each offer unique advantages in terms of generalization and accuracy. Therefore, applying the appropriate model based on the given situation of the problem is crucial for achieving effective sub-environment recognition in agents.

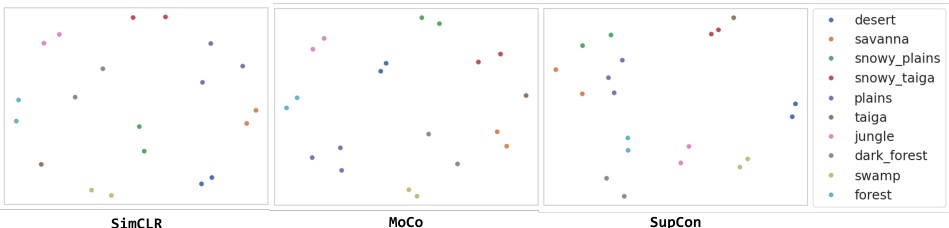

Figure 19: MDS visualization of biomes in Miniature environment. Jensen-Shannon divergence (JSD) is used as the measure between sub-environments. In SSL models, metrics are similarly distant across biomes, while in SupCon, distinct metrics reflect the latent class distribution of objects.

## D Chained Inference of OBSER Framework in Photorealistic Environment (Replica)

### D.1 Problem definition

In this section, we discuss the application of OBSER framework for navigation tasks. Suppose that a navigation task is defined as locating to the position $p_{\mathcal{A}}^q$ of a given object $x_q$ in a given environment $\mathcal{E} := \{(\mu_s, \mathcal{R}_s)\}_{s=1}^S$. Then an agent should infer i) the most probable sub-environment in episodic memory, ii) locate the most similar sub-environment with given memory and iii) find the object in such sub-environment.

Let an episodic memory $\mathcal{M} := \{(\hat{\mu}_m, \hat{\mathcal{R}}_m)\}_{m=1}^M$ be a set of observations $\hat{\mu}_m := \{x_{mo}\}_{o=1}^O$ and its locations $\hat{\mathcal{R}}_m := \{p_{mo}\}_{o=1}^O$. Depending on assumptions of the tasks, the location may be unknown or useless (unseen). With a given query $x_q$, the most probable sub-environment in episodic memory can be inferred with **object existence probability (object-environment)**:

$$i) \quad m^* = \operatorname*{arg\,max}_{m \in \{1, \cdots, M\}} \mathbb{E}_{x' \sim \hat{\mu}_m} \left[ \mathcal{K}_{\mathcal{Z}}(x_q, x'; \mathcal{F}) \right].$$

With a reachable region $\mathcal{N}_{\mathcal{R}}(p_{\mathcal{A}}; \mathcal{E}) := \{s | \mathcal{R}_s \subseteq \text{reachable}(p_{\mathcal{A}}, \mathcal{E}), s \in \{1, \cdots, S\}\}$ with the location of agent $p_{\mathcal{A}}$, an agent can retrieve the most similar sub-environment which minimizes **the KL divergence (environment-environment)** with given memory $(\hat{\mu}_{m^*}, \cdot)$.

$$ii) \quad s^* = \operatorname*{arg\,min}_{s \in \mathcal{N}_{\mathcal{R}}(p_{\mathcal{A}}; \mathcal{E})} \hat{D}_{\mathrm{KL}}(\hat{\mu}_{m^*} || \mu_s; \mathcal{F}).$$

After the agent reaches to $\mathcal{R}_{s^*}$, it explores the region $\mathcal{R}_{s^*}$ to find a target position which has **the same object with given query** $x^q$ **(object-object)**:

$$iii) \quad p_{\mathcal{A}}^q = \operatorname*{arg\,max}_{p_{\mathcal{A}} \in \mathcal{R}_{s^*}} \mathcal{K}_{\mathcal{Z}}(x_{p_{\mathcal{A}}}, x_q; \mathcal{F}),$$

with observation $x_{p_{\mathcal{A}}}^q$ in position $p_{\mathcal{A}}^q$.

## D.2 REPLICA ENVIRONMENT

We conduct experiments with Replica environment. We choose Replica environment since it is an high-quality indoor environment. By following the three consecutive step inferences, each models should retrieve appropriate object with the given queries. Figure 20 shows the overview of the procedure of the chained inferences.

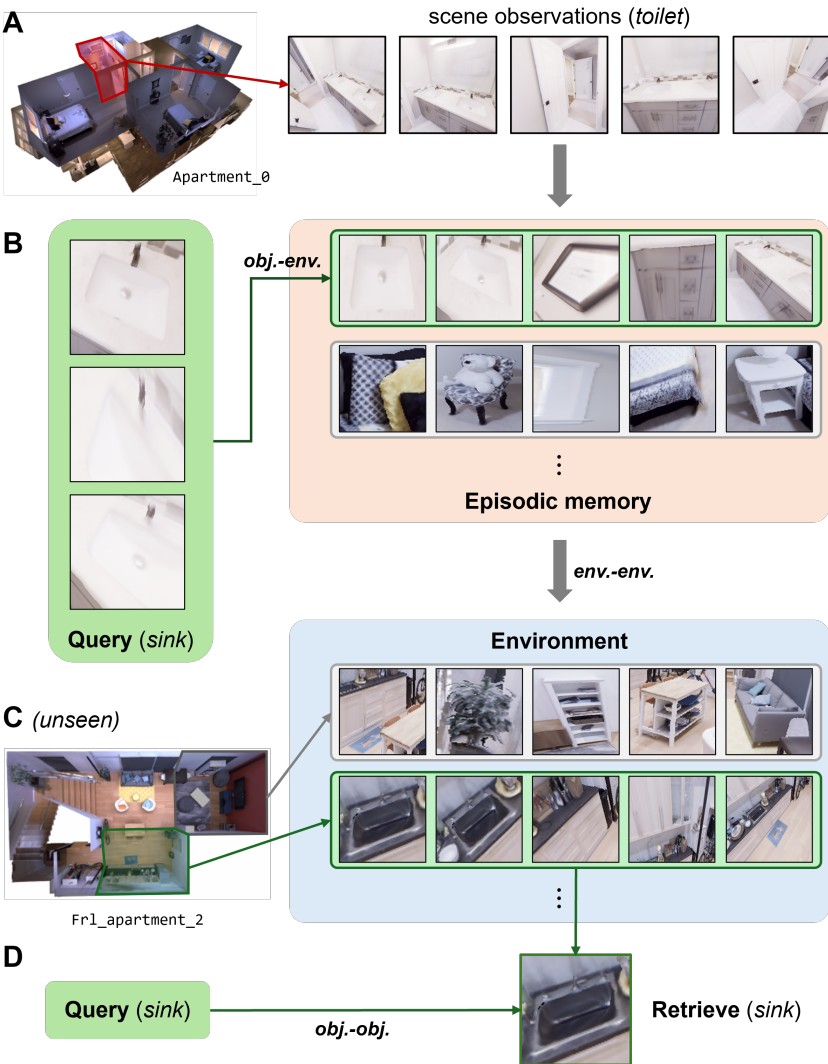

Figure 20: Visualization of the object retrieval task with the OBSER framework via three-step chained inference. A) An episodic memory is formed with object observations extracted from the scene observations of each sub-environment. B) When given a query, the agent first retrieves the most probable sub-environment from its episodic memory. C) With the retrieved memory, the agent finds the most similar sub-environment, D) and finds the most similar object in the sub-environment.

To focus on evaluation of the proposed framework, we utilize the gathered observations as both episodic memory and environment. We collect 960 random scenes from 48 rooms and extract object observations from each scene. We conduct experiments with two conditions: *seen* condition $\mathcal{M} = \mathcal{E}$ and *unseen* condition $(\bigcup_s \hat{\mathcal{R}}_s) \cap (\bigcup_m \hat{\mathcal{R}}_m) = \emptyset$. In *unseen* condition, we set the episodic memory only rooms with apartment_0 (consists of 13 rooms) and the environment with others. With 10 different objects as a query, we compute the accuracy of the inference for each model.

# E    REPRODUCIBILITY STATEMENT

To validate the proposed method with the ImageNet dataset, we use pretrained weights, which are provided publicly via GitHub. We also specify the seeds which are used for repeated experiments in the source code. For the Minecraft environment, we plan to upload the dataset, codes for data acquisition, and the Miniature map. The trained weights used for the experiments will also be uploaded.

