# OpenReview forum: "Object-Based Sub-Environment Recognition"
_ICLR.cc/2025/Conference — Submitted to ICLR 2025_

### Official Review · Reviewer_8Td2 · 2024-11-04

**Soundness:** 2
**Presentation:** 2
**Contribution:** 2
**Rating:** 6
**Confidence:** 1

**Summary:**

This paper proposes the Object-Based Sub-Environment Recognition (OBSER) framework. OBSER identifies sub-environments with three relationships: object-object, object-environment, and environment-environment relationship. The effectiveness of OBSER is measured with the proposed statistically separable (EDS) function in the Minecraft environment.

**Strengths:**

- OBSER identifies sub-environments with three relationships between objects and environments, and exhibits better distinguishability in terms of EDS compared to other off-the-shelf vision models.

**Weaknesses:**

- The application of OBSER is not clear. I'm not sure how OBSER will facilitate downstream tasks, e.g. decision agents in Minecraft like DreamerV3[1], Voyager [2], or GITM [3].

[1] Mastering Diverse Domains through World Models

[2] An Open-Ended Embodied Agent with Large Language Models

[3] Generally Capable Agents for Open-World environments via Large Language Models with Text-based Knowledge and Memory

**Questions:**

- What if the category of possible objects is unknown, e.g. in the open-set setting?

---

> ### Author Response · Authors · 2024-11-21
> **Response to Reviewer 8Td2 (1/1)**
>
> Dear Reviewer 8Td2,
>
> We are thoroughly thankful for your positive and insightful feedback. Here is our response to your comments.
>
> “The application of OBSER is not clear.” : To show the applicability of our framework in a more realistic environment, we additionally conducted experiments with a Replica environment, which is a 3D indoor environment. In this environment, we build chained inference, which contains all three relationships with the OBSER framework. The object retrieval task, which can be interpreted as visual navigation, is used for evaluation. When the object query is given, the framework first finds the most relevant room in its memory and finds the most similar room in an environment with the memory. The framework retrieves the object most similar to the given query. As a result, we found that the proposed framework can successfully retrieve the result by exploring a limited number of relevant rooms (1 or 3), which is much smaller than the total number of rooms (48 or 35). This implies that sub-environment recognition can be effectively applied to agents for efficient inferences.
>
> “What if the category of possible objects is unknown” : In our experiments, the framework didn’t have access to any category information from the environment. Our method is based on metric learning (and SSL), so the framework doesn’t require explicit class information.
>
> However, when the problem is open-set, which contains unseen objects that are not used for training, metric learning-based models, such as SupCon, may fail in some situations. In metric learning, the model is trained with positive pairs (and negatives) of the data which is selected with explicit class. However, SSL models can be robust in such environments because they are trained only with inductive biases in the data domain. In the Replica environment, we conducted with an “unseen” setting, and in this situation, the framework can only be accessible to rooms in “apartment_0” for episodic memory. In this setting, SupCon underperforms MoCov3 and DINOs, and we think this phenomenon occurs for this reason.

---

### Official Review · Reviewer_WjYy · 2024-11-04

**Soundness:** 3
**Presentation:** 2
**Contribution:** 2
**Rating:** 5
**Confidence:** 2

**Summary:**

The paper proposes the Object-Based Sub-Environment Recognition (OBSER) framework, a novel Bayesian framework for measuring object-environment and environment-environment relationships using a feature extractor trained with metric learning. The key idea is the introduction of a statistically separable (EDS) function and using it to perform (i) object-object similarity, which involves obtaining the closest class of objects from a list given a query object, (ii) object-environment recognition, which involves retrieving the closest environment to a given object and (iii) environment-environment recognition, which defines the difference between two sub-environments. Experiments to recognize environments are done on two datasets, the ImageNet based dataset, and a dataset of curated environments from Minecraft.

**Strengths:**

-	The primary motivation behind the paper is sound. Indeed, environment recognition using relationships between objects and environments is an interesting problem in embodied agents.
-	The results do illustrate the claim that higher difference between epsilon and delta values lead to a better accuracy score. This is reflected in models for both Tables 1 and 2.

**Weaknesses:**

-	Not clear what objects and environment are: By reading the paper starting from the introduction, it is not clear what objects and environments mean. The authors show some examples of objects and biomes as environment in the Minecraft example, but none for the ImageNet dataset. This makes it difficult to understand the contribution of the work.
-	Results are difficult to interpret: There are mainly just two results described in the paper in Tables 1 and 2, which are the respective classification accuracies for ImageNet and Minecraft datasets. It is difficult to interpret these tables. For instance, are the differences between metric and self-supervised learning methods the main observation, or the relationship between EDS values and different models?
-	Real-world applications are unclear: While the motivation of the work is sound, it is not straightforward to interpret how this paper contributes to real-world object and scene recognition. The paper does not contain any examples of real-world object scenes or object recognition. The only examples provided are for Minecraft, which is still a simulation environment, and not reflective of human-centric objects and environments.
-	[Minor] Unnecessary math: There is a lot of mathematical terminology introduced in Sections 3-5 (besides the main contribution in Section 4.1), which are not necessary to be present in the main paper.
-	[Minor] Many details are missing from the main paper and are present in the supplementary. The authors should consider transferring some details about the implementation from supplementary to the main paper. For any details, the reader has to constantly switch between paper and supplementary, which is not a good user-experience.

**Questions:**

-	Is there any justification for the type of classifiers provided in Tables 1 and 2? In Table 1, linear, mean and KNN classifiers are used, while only mean and KNN classifiers are used in Table 2. Also, why the difference between the shots of mean vs KNN (1,3,5 vs 3,5,7).
-	For the ImageNet dataset, what are some examples of objects and environments? It is unclear from reading the paper.
-	In conclusion, the authors mention that by integrating the proposed method with embodied object recognition or navigation modules, inference accuracy can be improved. Can the authors provide some justification with a real-world use-case about what is the intuition behind this?
-	What classification accuracies are mentioned in Tables 1 and 2? Is it the object-object similar classes?

**Details Of Ethics Concerns:**

No ethical concerns observed.

---

> ### Author Response · Authors · 2024-11-21
> **Response to Reviewer WjYy (1/1)**
>
> Dear Reviewer WjYy,
>
> We appreciate your recognition of the motivation behind our work and the validity of the results supporting our claims. Here are our responses to your thoughtful comments.
>
> “some examples of objects and environments in ImageNet” : Conceptually, a sub-environment with ImageNet can be interpreted as separated “room” or “place”, and an object as “observed target”. We validated the proposed framework with a subset of ImageNet with different data distributions. With the following steps, we generated these artificial environments.
>
> - First, we randomly selected 10 (or 40) classes in ImageNet for each seed.
> - With a given class distribution (rho), We sampled data of each class from the dataset.
> - We analyzed the framework with gathered data as empirical distribution.
>
> Also, to validate our framework in a more realistic environment, we additionally conducted experiments with the Replica environment, which is a 3D indoor environment. In this case, a sub-environment becomes a room, and an agent can observe objects in each room.
>
> “mainly just two results” : Since we visualized that all three relationships (object-object, object-environment, and environment-environment) can be estimated with the OBSER framework in the forms of tables (1,2), graphs (5,6), and qualitative results with demos (7-9), we cannot agree that the main results are just two.
>
> “For instance,… different models?”, “any justification for the type of classifiers provided in Tables 1 and 2” : Tables 1 and 2 show the relationship between EDS values and classification accuracy because the classification task highly depends on the object-object relationship. For classification tasks, we chose the KNN classifier and mean classifier to show the influence of concentration and separability. We chose mean classifier rather than linear probing because mean classifier can be used without explicit label information (this means we can compute mean classifier accuracy directly with any set of labels). If the representations are not separated, both KNN and mean classification accuracy are low. If the representations are separated but not concentrated, then KNN accuracy is low, while mean classification accuracy is high. Also, using top-1 accuracy with the KNN classifier is often noisy (because it only considers the nearest neighbor), so we used (3, 5, 7) instead of (1, 3, 5).
>
> “Real-world applications are unclear”, “some justification with a real-world use-case about what is the intuition behind this” : As mentioned above, We conducted additional experiments with photorealistic environment. We chose the object retrieval task in multiple sub-environments (in this case, rooms), which can be interpreted as visual navigation. We design chained inference with all three relationships. With a given query, the framework first retrieves the most relevant room in its memory. With the memory, the framework finds the most similar room and finds the queried object. As a result, we found that our framework can find the queried object by searching with a limited number of relevant rooms (1 or 3) rather than iterating all rooms (48 or 35) for exploration.
>
> “Unnecessary math” : To make sure that a proposed method is correct, we think that theoretical support to guarantee its correctness is essential. For this reason, we introduced the EDS function to show that the precision of inference with the framework is guaranteed with an optimized feature extractor with low epsilon and high delta. So, we argue that our theoretical attempt is necessary and justified.
>
> Finally, we have moved several contents in the main paper and the appendix for more readability.

---

> > ### Comment · Reviewer_WjYy · 2024-12-03
> >
> > The authors have partially addressed my concerns by providing some examples from the Replica dataset. While I do appreciate the effort put forth by the authors, I still feel that the real-world applications of this work is limited, as noted by other reviewers also. For instance the results of metric learning model SupCon appear to be one of the best for ImageNet (another realistic dataset) and Minecraft, but amongst the worst for Replica (at least on the unseen case). This makes it difficult to root for which backbone best benefits from the OBSER framework.
> >
> > Based on the author responses, and looking at the other reviews, I will raise my rating to 5.

---

> > > ### Author Response · Authors · 2024-12-03
> > > **Response to Reviewer WjYy (Metric learning and SSL)**
> > >
> > > Dear Reviewer WjYy,
> > >
> > > We sincerely thank you for recognizing our effort to enhance the novelty of the OBSER framework through its application to a photo-realistic (real-world-like) environment. We would like to share our perspective regarding the concerns you have raised.
> > >
> > > - As mentioned in LN 522, we observed that SSL models perform more robustly than metric learning model (SupCon) in the Replica environment, especially in unseen settings. In Appendix C.3.2 and C.4, we also observed similar results with the Minecraft environment. We claim that in this phenomenon, self-supervised learning methodologies enable representations to encompass more information than the class information defined by metric learning, thereby facilitating robust sub-environment recognition in cases where data is less structured or newly occurred (unseen).
> > > - We built episodic memory of sub-environments differently for experiments with each environment. We designed the ImageNet experiment to show the relationship between EDS values and measures of sub-environment recognition. Therefore, in ImageNet, we sampled data from the dataset using arbitrary class distributions.
> > > - On the other hand, in Minecraft and Replica, we gathered data with ego-centric observations to demonstrate the real-world agents: we first divided the environment into sub-environments, gathered scene observation, and extracted object observations to form episodic memory. The gathered data often contains ambiguities, such as obscured observations and ambiguous objects, and dealing with such ambiguities is important to achieve real-world agents. We claim that the results of Minecraft and Replica experiments are more essential than the results of ImageNet in showing the applicability of the OBSER framework to the real world.
> > > - In summary, metric learning models enable numerically more precise inference in well-defined data and environments but may face difficulties in generalizing to less-structured data and environments. In contrast, SSL models may produce relatively noisy measurements for the inference but demonstrate greater strengths in generalization (LN 1628-1635, Appendix C.4). We claim that using the SSL-based OBSER framework is more effective than metric learning-based models for achieving real-world agents.
> > >
> > > To ensure that our observation is effectively conveyed to the readers, we will incorporate more details about it in Section 6 (Experiments) and Section 7 (Conclusion). We hope this sufficiently addresses your concerns and helps readers better understand our work.
> > >
> > > Once again, we deeply appreciate your interest and commitment to our work.
> > >
> > > Best regards,
> > >
> > > Authors of Submission 6783

---

### Official Review · Reviewer_9vYe · 2024-11-06

**Soundness:** 2
**Presentation:** 2
**Contribution:** 2
**Rating:** 5
**Confidence:** 2

**Summary:**

The paper presents a Bayesian framework to recognize sub-environments within complex, dynamic environments. OBSER enables agents, like robots, to identify sub-environments based on objects present, facilitating task-driven navigation and inference in open-world scenarios. The framework introduces EDS function to improve the robustness of feature representations and utilizes metric learning for object-object, object-environment, and environment-environment relationships.

**Strengths:**

OBSER provides a holistic approach to sub-environment recognition, measuring three relationships—object-object, object-environment, and environment-environment—which enables better contextual awareness.

The introduction of the EDS function to assess separability and concentration offers a robust way to manage feature representations, addressing a gap in object-based environmental recognition.

**Weaknesses:**

I find that this method may be challenging to implement for embodied robots. First, constructing episodic memory seems crucial for task completion success, yet several questions arise: (1) How was this memory constructed? (2) How could it be constructed effectively with limited experience? (3) How can retrieval be managed efficiently as memory size increases?

Most importantly, I am uncertain about how the object-object, object-environment, and environment-environment relationships contribute to embodied tasks. Without ablation studies or proof, it’s hard to determine the critical importance of these relationships.

Are there other baselines with which EDS could be compared? The paper would benefit from broader comparisons with other state-of-the-art environment recognition frameworks to better highlight OBSER's distinct advantages and limitations in context.

The diversities of Minecraft environment and objects seems limited.

OBSER's reliance on object distribution might limit its effectiveness in sub-environments where objects are scarce or ambiguous, which could impact performance in less structured real-world spaces.

**Questions:**

Please refer to the weakness.

---

> ### Author Response · Authors · 2024-11-21
> **Response to Reviewer 9vYe (1/1)**
>
> Dear Reviewer 9vYe,
>
> We appreciate that you highlighted our intention of introducing the OBSER framework and EDS function. We are also thankful for the insightful comments about applying our framework to the embodied agents. Here is our response.
>
> “I find that this method may be challenging to implement for embodied robots”: In this paper, we focus on the inference of the sub-environment recognition, so we assumed that the framework can work independently with embodiments when it can reach the observations. For the Minecraft experiment, we deployed randomly exploring agents to gather ego-centric observations. In the Replica experiment, we randomly gathered scene observations. We think both are commonly used methods for gathering environmental information with embodied agents.
>
> “limited, or excessive experience” : Our framework uses self-supervised learning models (and metric learning) and data augmentation can be used to enhance the sample efficiency. This method has enhanced performance in offline RL [1]. Also, our method uses representation, not data itself, and time complexity (and also space complexity) only takes O((M+N)^2*D) with memory size M, N, and dimension D for KL divergence estimation. Even when the experience is huge to harm the memory or computation cost, we can reduce unnecessary (or duplicated) data from the memory.
>
> “how the object-object, object-environment, and environment-environment relationships contribute to embodied tasks” : To show the applicability of the OBSER framework in embodied tasks, we validate the chained inference of the OBSER framework for object retrieval tasks, such as visual navigation. We use the Replica environment and set each room as a sub-environment. As a result, we found that retrieval with a small number of relevant rooms (top-1 or top-3) can bring a similar performance to one with all rooms (48 or 35). This indicates that sub-environment recognition can enhance the efficiency of both inference and exploration.
>
> “The diversities of Minecraft environment and objects seems limited.” : In the Minecraft environment, we chose common biomes in “overworld,” the main map of the game. There are some similar biomes that contain the same objects. However, this makes it harder to infer the sub-environments accurately. The proposed framework can successfully detect such minor differences (details) between biomes.
>
> “objects are scarce or ambiguous, … in less structured real-world space” : As mentioned in the above responses, the proposed framework also shows robustness and efficiency with the object retrieval task in photo-realistic environments. For this experiment, we used randomly gathered observations which are less structured and contain ambiguous data, instead of curated ones.
>
> [1] Schwarzer, Max, et al. "Data-efficient reinforcement learning with self-predictive representations." *arXiv preprint arXiv:2007.05929* (2020).

---

### Official Review · Reviewer_d5mr · 2024-11-07

**Soundness:** 2
**Presentation:** 3
**Contribution:** 2
**Rating:** 6
**Confidence:** 2

**Summary:**

The authors propose a framework for an agent to infer its sub-environment through the measurements of object-object, object-environment, and environment-environment relationships. The authors validate their framework in Minecraft, while also providing some preliminary results on the ImageNet dataset. The paper is well structured, and the authors show several relevant results, including the relevance of statistically separable EDS functions to achieve accurate measures for their downstream environment inference.

**Strengths:**

The paper is generally well structured. The authors explain each part of their proposed method in detail, including for example the relevance of statistically separable EDS functions to achieve accurate measures for their downstream environment inference, and the empirical implication of hyperparameter choice for downstream inference (e.g. the choice of Tau for KL divergence in Figure 6.).

**Weaknesses:**

It is hard to quickly have a notion of which parts of the proposed methods, exactly, are novel. The authors use several existing methodologies in their proposed framework, but fail to appropriately specify which of these, in particular, are novel propositions or implications. It is also hard to connect the motivation of this work to the tasks and results shown. In particular, the emphasis on the motivation for "real-world" applications, and complex natural environments is lost by the simplicity of the test settings (e.g. virtual world or fixed datasets).

**Questions:**

- It would be worthwhile to adjust the tone of the claims in the paper to better align with the results shown. The results may show interesting results in a "simulated environment, towards more complex environmental settings" perhaps even eventually leading to real-world, but as far as this work goes there is a wide gap between simulated and real-world settings, since no robotic experiments were provided. Below are some of the most relevant parts, strongly suggesting (non-existing) results in real-world settings
     - The abstract
     - The introduction should reflect this (3rd claim)
     - Figure 1: The caption should be updated (it is not, in fact, a real-world agent)
     - Title in 6.2 should change

- Section 4: Which of these are new propositions and which of these are derived from existing work? This should be made very explicit.
- Missing y-label in Fig. 5 and Fig. 6
- English should be improved throughout:
  e.g. "which computes the kernel density accumulated with class-wise distribution.", or "We utilized pretrained weights for every models." etc.

---

> ### Author Response · Authors · 2024-11-21
> **Response to Reviewer d5mr (1/1)**
>
> Dear Reviewer d5mr,
>
> We are thoroughly thankful for your thoughtful and positive feedback. We appreciate that you found the structure of the paper clear and appreciated the detailed explanations of our methodology, including the statistical and empirical aspects of our approach. Here is our response to your review.
>
>
> "which parts of the proposed methods, exactly, are novel". "connect the motivation of this work to the tasks and results shown": Our main goal of this work is introducing sub-environment recognition with three fundamental relationships and suggesting metric learning (or SSL) based approaches to estimate these relationships. To make sure the proposed framework can be applied to solve tasks in a realistic environment, we conduct additional experiments with object-retrieval tasks in a Replica environment. We found that with sub-environment recognition, an agent can more efficiently explore by prioritizing the order of exploration from relevant sub-environment to irrelevant ones.
>
>
> "complex natural environments is lost by the simplicity of the test settings": We choose Replica environment as an additional environment because the environment is photorealistic environment with high quality 3D meshes (MP3D environment is much larger, but somewhat noisy). We gather ego-centric scene observations to extract object observations for episodic memories, making the problem harder because the data is less structured. However, as a result of the experiments, we found that our proposed framework can reliably infer in such environments. For more details, please refer to Section 6.2 and Appendix D.
>
>
> "adjust the tone of the claims": We have found that some expressions you mentioned might reduce the clarity of our claims. Thus, we have adjusted them to focus on our main research topic. You can find the difference in the abstract, introduction, several captions, and conclusion.
>
>
> "Section 4: Which of these are new propositions ": We newly define the two important properties of representations (epsilon and delta) in kernel densities. We also "rederive" the optimization of the EDS function, minimizing the KL divergence, to find an upper bound in terms of epsilon and delta. With this upper bound, we can guarantee the optimization of such loss can bring the optimization of the EDS function. In previous works [1,2], the authors derive the InfoNCE from classifier or contrastive learning, which is different from distribution matching. Optimization with KL divergence is introduced in [3], but they used an alternative way to derive the optimization problem.
>
>
> We also have added a y-axis for several figures and modified expressions for more readability.
>
>
> [1] Oord, Aaron van den, Yazhe Li, and Oriol Vinyals. "Representation learning with contrastive predictive coding." *arXiv preprint arXiv:1807.03748* (2018).
>
> [2] Khosla, Prannay, et al. "Supervised contrastive learning." *Advances in neural information processing systems* 33 (2020): 18661-18673.
>
> [3] Choi, Won-Seok, et al. "DUEL: Duplicate Elimination on Active Memory for Self-Supervised Class-Imbalanced Learning." *Proceedings of the AAAI Conference on Artificial Intelligence*. Vol. 38. No. 10. 2024.

---

> > ### Comment · Reviewer_d5mr · 2024-11-25
> >
> > The authors mostly address my concerns.
> >
> > As a minor comment, Figure 1 is still not very expressive, and the caption is not very informative or obvious given the figure.

---

> ### Author Response · Authors · 2024-11-26
> **Response to Reviewer d5mr (Modification of Figure 1)**
>
> Dear Reviewer d5mr,
>
> We are very pleased that our explanations and revisions have addressed your concerns. Your comment has been immensely helpful in improving the clarity and novelty of our work.
>
> As you mentioned, we have modified Figure 1 and its caption to include more information about the task and inference with sub-environment recognition. Please refer to Figure 1 in the latest version of the paper to ensure that your concern is fully addressed.
>
> If you have further suggestions, please let us know during the discussion phase.
>
> Best regards,
>
> Authors of submission 6783

---

### Author Response · Authors · 2024-11-21
**Response to all reviewers**

Dear Reviewers,

First, we would like to express our gratitude for the positive consideration of our work. The reviewers expressed that the motivation of our work is sound (9vYe, WjYy), the paper is well-structured (d5mr), and the experiments support our claims well (d5mr, 9vYe, WjYy). We also appreciate the insightful comments to improve our paper. To announce important changes in our paper, we share this information in a separate comment instead of replying to each comment.

Our research topic is to propose a new sub-environment recognition framework with the metric learning-based method. We designed the sub-environment recognition with three fundamental relationships (object-object, object-environment, and environment-environment). In this work, we focus on expressing and validating such relationships via kernel density estimation with a feature extractor trained with metric learning and SSL, rather than solving embodied tasks with these relationships. We believe that combining our framework with embodied agents is a promising research topic. For this reason, we have found that several expressions in the previous revision can be read unintendedly, which may lead to a misunderstanding of the focus of our paper. To clarify our claims, we adjust expressions such as "embodied agent" throughout the paper (especially in Introduction and Conclusion).

Nevertheless, we claim that our method can be applied to real-world situations, so we perform additional experiments with Replica, which is a 3D photorealistic indoor environment introduced in [1]. In these experiments, we designed chained inference with all three relationships. We set each room of each environment as the sub-environments and apply the OBSER framework to solve the object retrieval tasks in multiple sub-environments. As a result, we found that our method can reliably retrieve the queried objects with only a limited number of relevant rooms (1 and 3) for search, which is much smaller than the total number of rooms (48 in the seen setting and 35 in the unseen setting). This suggests that with sub-environment recognition, the agent can efficiently retrieve the goal by reducing unnecessary exploration of irrelevant sub-environments. Also, we found that the metric learning model (SupCon) performs worse than SSL models in unseen settings, which implies that SSL models are robust in less-structed (curated) but more general environments. Please refer to Section 6.2 and Appendix D in the revised version for more details.

[1] Straub, Julian, et al. "The Replica dataset: A digital replica of indoor spaces." *arXiv preprint arXiv:1906.05797* (2019).

---

### Meta-Review · Area_Chair_GJtW · 2024-12-17

**Metareview:**

The paper received borderline ratings (6,6,5,5). The reviewers identified several weaknesses, such as challenges in pinpointing novel aspects, limited diversity in Minecraft environments, unclear real-world applications, and a lack of clarity in the definitions of objects and environments. The author provided responses to the reviewers. The AC checked the paper, the reviewers and the responses. The AC did not find the responses convincing enough. For example, the AC is in agreement with reviewer WjYy that the real world applications are limited. Also, the authors did not adequately address the questions posed by Reviewer 9vYe “(1) How was this memory constructed? (2) How could it be constructed effectively with limited experience? (3) How can retrieval be managed efficiently as memory size increases?”. Due to these issues, the AC recommends rejection.

**Additional Comments On Reviewer Discussion:**

Reviewer d5mr mentioned that their concerns were addressed by the rebuttal. Reviewer WjYy appreciated the new experiments but they still had concerns regarding the real world applicability. The AC checked the follow up response by the authors but did not find it convincing enough. The AC checked the responses to reviewer 9vYe and 8Td2, but could not find good answers to some of the questions (some examples mentioned above). Therefore, the AC decided that the paper needs major revision and recommended rejection.

---

### Decision · Program_Chairs · 2025-01-22

Reject